# A Closer Look at the Application of Causal Inference in Graph Representation Learning

## Abstract

Modeling causal relationships in graph representation learning remains a fundamental challenge. Existing approaches often draw on theories and methods from causal inference to identify causal subgraphs or mitigate confounders. However, due to the inherent complexity of graph-structured data, these approaches frequently aggregate diverse graph elements into single causal variables—an operation that risks violating the core assumptions of causal inference. In this work, we prove that such aggregation compromises causal validity. Building on this conclusion, we propose a theoretical model grounded in the smallest indivisible units of graph data to ensure that the causal validity is guaranteed. With this model, we further analyze the costs of achieving precise causal modeling in graph representation learning and identify the conditions under which the problem can be simplified. To empirically support our theory, we construct a controllable synthetic dataset that reflects real-world causal structures and conduct extensive experiments for validation. Finally, we develop a causal modeling enhancement module that can be seamlessly integrated into existing graph learning pipelines, and we demonstrate its effectiveness through comprehensive comparative experiments. *Code and data can be found in the supplementary materials.*

## 1 Introduction

In deep learning, accurately modeling causal relationships is a key step toward building trustworthy AI (Li et al., 2023). Causal relationships represent the real cause-and-effect links between variables, possessing deterministic certainty, unlike probabilistic correlations that may involve false connections. This is especially important for graph representation learning using neural networks (Gao et al., 2023), as the complex connections between nodes and the built-in structure of graph data are likely to create confusing biases and false correlations (Jukna, 2006). In common applications, like recommendation systems (Wu et al., 2022a), drug discovery (Takigawa & Mamitsuka, 2013), and social network analysis (Tan et al., 2019), these biases often appear as the challenge of separating key causal factors—like telling popularity from true preference, bias from biological function, and similarity from influence. Additionally, since graph models are often used to study systems with spreading effects, predictions based on such false correlations are not only likely to fail, but their negative outcomes can also be made much worse when spread by the network structure.

In recent years, researchers have sought to address this issue (Wu et al., 2022b; Fan et al., 2022; Gao et al., 2023; 2024; Sun et al., 2025; Zhao et al., 2025). Mirroring trends in other domains of neural networks (Kaddour et al., 2022), they have begun to integrate principles of causal inference into graph representation learning, achieving notable success. These approaches typically function by either identifying causal subgraphs within the graph data or eliminating confounders—extraneous variables that interfere with the accurate modeling of causal relationships. The efficacy of these methods has been validated on both synthetic datasets designed for causal benchmarks and real-world datasets.

However, the aforementioned methods often merge multiple graph components—such as nodes and edges—into a single causal variable in their analysis. For example, they typically consider the entire causal subgraph or confounder as one unified variable. In an ideal scenario, if the causal relationships between the variables in the studied graph data align with the variables created by these methods, then, based on the theory of Spirtes (2009), such an analysis poses no issues and satisfies the prerequisites for causal inference applications. However, in real-world situations, the complex interrelationships within graph data lead to highly intricate interactions between the variables, which do not meet this ideal condition. We provide an intuitive example in Figure 1. This practice raises an important question: **what impact does such merging of variables have on the granularity and accuracy of causal analysis in graph representation learning?** Our findings show that such a simplification inevitably violates the two fundamental premises of causal inference, making it inapplicable. Please refer to Proposition 1 for details. Consequently, a new question emerges: **from a causal theory perspective, is it possible to achieve perfectly accurate causal relationship modeling in graph representation learning, and at what cost?**

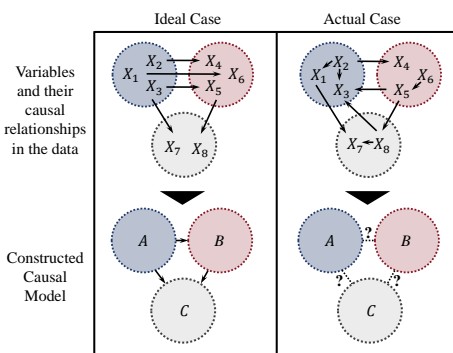

Figure 1: An example of the ideal case and the actual case in causal model building. In the actual case, the causal model cannot be constructed as in the ideal case due to the complex reciprocal causal relationships between the merged variables.

In this paper, we address the aforementioned issues from a rigorous theoretical perspective, supported by sufficient experimental evidence. Specifically, we develop a new theoretical model for studying causal modeling in graph representation learning. Such a model is built upon the smallest divisible variables in graph data, ensuring strict adherence to the fundamental theoretical premises of causal inference. Based on this model, we conduct a series of analyses and proofs. Subsequently, we carry out experimental analyses, constructing an artificial synthetic dataset that closely resembles real-world scenarios and performing multiple experiments and analyses. Building upon the aforementioned research outcomes, we also develop a new plug-and-play causal modeling enhancement module. Our contributions can be summarized as follows:

- We proposed a new theoretical model that strictly adheres to the fundamental premises of causal inference for studying causal modeling in graph representation learning.

- Based on the proposed model, we derive and prove a series of theories concerning causal relationship modeling in graph representation learning, including its costs and simplifications.

- We constructed a synthetic graph dataset with controllable causal relationships, more closely resembling real-world scenarios, for causal relationship modeling research. Additionally, we conducted a series of experiments for cross-validation with the proposed theory.

- Based on the aforementioned research and conclusions, we introduced a novel plug-and-play module for optimizing causal relationship modeling in graph representation learning.

## 2  RELATED WORKS

### 2.1  CAUSAL LEARNING

Causal learning focuses on identifying and modeling causal relationships rather than mere correlations, utilizing methods like causal graph models (Kocaoglu et al., 2019), causal inference (Pearl & Mackenzie, 2018), and causal discovery (Zheng et al., 2018). Recent advances integrate causal learning with neural networks to enhance interpretability and generalization by incorporating causal structures (Chattopadhyay

et al., 2019; Zhang et al., 2020), removing spurious correlations (Zhang et al., 2021), and analyzing latent variable relationships (Yao et al., 2024). Causal deep generative models disentangle factors like object shape and texture for counterfactual generation (Sauer & Geiger, 2021), while causal reinforcement learning improves robustness through stable representations (Zhang et al., 2024). These approaches advance causal understanding in neural networks across prediction, generation, and decision-making tasks (He et al., 2023; Bagi et al., 2023; Cheng et al., 2024; Zeng et al., 2023; Zhu et al., 2023; Yu et al., 2024).

## 2.2 CAUSAL RELATIONSHIP MODELING WITH GRAPH NEURAL NETWORKS

Causality plays a crucial role in addressing the complexity of graph data (Lippe et al., 2023; Sui et al., 2022), especially in fields such as finance (Wang et al., 2022), medicine (Shang et al., 2019), and biology (Zitnik et al., 2018). Most current research focuses on how to enable Graph Neural Networks (GNNs) to model causal relationships in graph representation learning. These studies typically employ two main approaches: (1) modeling causal subgraphs, where Fan et al. (2024) proposed a causal representation framework for stable GNNs, Wu et al. (2022b) developed a discovering invariant rationale (DIR) strategy, and Chen et al. (2022) introduced Causal-Inspired Invariant Graph Learning (CIGA) for OOD generalization; and (2) eliminating confounding factors, where Fan et al. (2022) proposed a decoupled GNN framework, Gao et al. (2024) designed a lightweight optimization module, and Wu et al. (2024) employed causal inference-inspired learning to overcome confounding biases. Both of these approaches adopt variable merging in their causal analysis, and our work aims to investigate the impact of this merging from both theoretical and experimental perspectives.

## 3 THEORETICAL ANALYSIS

### 3.1 BASIC MODEL

As discussed above, research on graph representation learning from a causal perspective often merges numerous node and edge-level variables, making it difficult to ensure causal validity. Additionally, categorizing complex, interdependent variables within a graph dataset $\mathcal{G} = \{G_i\}_{i=1}^{|\mathcal{G}|}$ as "confounders" or "causal subgraphs" can impact the effectiveness of causal analysis methods, violating two key assumptions in causal inference: the Causal Markov Assumption and the Causal Faithfulness Assumption (Pearl, 2009). Formally, we propose the following proposition:

**Proposition 1** *When the variables within graph dataset $\mathcal{G}$ are merged to form a new and smaller variable set $S$, in certain cases, it becomes impossible to construct a causal model based on $S$ while still satisfying the two key prerequisites for applying causal inference methods—namely, the Causal Markov Assumption and the Causal Faithfulness Assumption.*

The proof can be found in **Appendix C.1**. This influence warrants more rigorous and systematic theoretical investigation. To study the problem, we formalize it using a Structural Causal Model (SCM) (Pearl, 2009), which serves as a framework for representing causal relationships among variables. An SCM consists of a set of variables and a corresponding set of relations that describe how each variable is causally influenced by others. In Figure 2, we illustrate the SCM as a Directed Acyclic Graph (DAG), in line with standard practices in causal inference. In this representation, vertices correspond to random variables, while edges denote the causal relationships between them. **The SCM we construct treats individual elements in the graph—such as nodes and edges—as separate variables, enabling an analysis that strictly adheres to the principles of causal theory.**

In Figure 2, let $U = \{U_i\}_{i=1}^{|U|}$ denote the set of exogenous variables, $X = \{X_i\}_{i=1}^{|X|}$ represent all the smallest divisible variables included in the graph, and $Y = \{Y_i\}_{i=1}^{|Y|}$ indicate the set of label variables. The

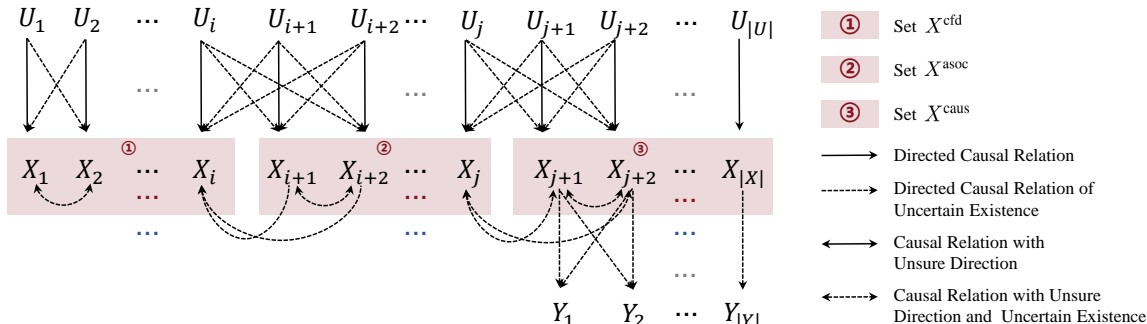

Figure 2: Graphical illustration of the proposed SCM of the graph representation learning scenario.

edges in the graph illustrate the causal relationships between the variables. For each variable $X_i$, there is a direct causal relationship with its corresponding exogenous variable $U_i$, and $U_i$ may also exert causal influences on other variables within $X$. These uncertain causal relationships are represented by dashed arrows. Furthermore, since we do not observe the values of the exogenous variables, our focus is solely on their causal influence on $X$. Consequently, all edges between $U$ and $X$ are directed from $U$ to $X$. Due to the large number of variables and edges in the graph, it is impractical to display each element individually. To address this, ellipses are employed to represent omitted variables and edges.

For the set $X$, we divide it into three subsets. As illustrated in Figure 2, subset one consists of variables that do not have causal paths to the label set $Y$ and may act as confounders. We denote this subset as $X^{\mathrm{cfd}}$. Subset two includes variables that do have paths to the labels $Y = \{Y_i\}_{i=1}^{|Y|}$ but are not part of $\bigcup_{i \in \{1,2,\ldots,k\}} Pa(Y_i)$, where $Pa(\cdot)$ represents the parent node set. Specifically, these variables are not parent nodes of any label $Y_i$, but they are causally associated with $Y$. We refer to this subset as $X^{\mathrm{asoc}}$. Subset three is $\bigcup_{i \in \{1,2,\ldots,k\}} Pa(Y_i)$, and we denote this subset as $X^{\mathrm{caus}}$. As illustrated in the figure, the causal relationships among variables are highly complex and uncertain. Within $X^{\mathrm{cfd}}$, $X^{\mathrm{asoc}}$, and $X^{\mathrm{caus}}$, variables may exhibit mutual causal associations. Moreover, variables in $X^{\mathrm{asoc}}$ may have causal connections with those in $X^{\mathrm{cfd}}$, while variables in $X^{\mathrm{asoc}}$ and $X^{\mathrm{caus}}$ may hold causal relationships in arbitrary directions. We demonstrate the validity of the SCM:

**Theorem 2** *The proposed SCM in Figure 2 can characterize the general causal relationships between various variables in the graph representation learning scenario. Furthermore, such an SCM satisfies the Causal Markov Assumption and the Causal Faithfulness Assumption.*

The proof can be found in **Appendix** C.1.

## 3.2 FURTHER DISCUSSION

Based on the proposed SCM that strictly satisfies the premises of causal inference, we wonder what it would cost to achieve perfectly accurate causal modeling using a GNN model? Intuitively, this would require analyzing every individual data element within the graph and conducting interventions. To this end, we conducted a theoretical analysis and, for both atomic interventions (intervening on a single variable at a time) and non-atomic interventions (intervening on multiple variables simultaneously), we derived the corresponding lower bounds on the number of interventions required:

**Theorem 3** *Based on the SCM in Theorem 2, when utilizing GNN to model causal relationships, for atomic interventions, the lower bound of the number of interventions required is*

$$\min_{\mathcal{M}^{micro}} \left( \left\lceil \frac{\frac{1}{\lambda} \left| \left( \bigcup_{i=1}^{|\mathcal{G}|} G_i \right) \right| + |Y| - r(\mathcal{M}^{micro})}{2} \right\rceil \right), \text{ where } \mathcal{M}^{micro} \text{ denotes any DAG that is equivalent to the}$$

graphical representation of the ground truth causal model, and the vertex set of $\mathcal{M}^{micro} = \left( \bigcup_{i=1}^{|\mathcal{G}|} G_i \right) \cup Y$, $\lambda$ denotes the average times that each variable occurs among each of the samples within dataset $\mathcal{G}$, $r(\cdot)$ calculates the total number of maximal cliques. For non-atomic interventions, the number of interventions required exceeds $\mathcal{O} \left( \min_k \left( \frac{\frac{1}{\lambda} \left| \left( \bigcup_{i=1}^{|\mathcal{G}|} G_i \right) \right| + |Y|}{k} log \left( log \left( k \right) \right) \right) \right)$.

The proof can be found in **Appendix** C.3. Theorem 3 provides a lower bound on the number of interventions required. Based on this theorem, for graph datasets, achieving accurate causal modeling would necessitate an extremely large number of interventions—at least on the order of $\mathcal{O} \left( \bigcup_{i=1}^{|\mathcal{G}|} G_i \right)$. Take the Citeseer dataset (Caragea et al., 2014) as an example; the required number of interventions would amount to several thousand. Given that interventions themselves are highly costly—and sometimes even infeasible—this raises a critical question: is it possible to achieve accurate causal modeling without performing such an excessive number of interventions?

**Theorem 4** *Assume there exists a GNN model that satisfies the infinite approximation theorem (Cybenko, 1989), and that interventions are applied to ensure the GNN models the causal relationships between the graph variables and the labels. In this case, when applying causal inference in graph representation learning, it is possible to merge some variables from the original set $X$ to form a new set $S$, where $|S| < |X|$, while ensuring that the causal relationships between the graph data and the labels are accurately modeled. However, the following conditions must be met:*

*(1) Variable $s$ in $S$ that satisfies $s \in Pa(Y)$ cannot simultaneously contain both the parent and child nodes of another variable $v \in X$.*

*(2) Variables within $X^{caus}$ cannot be merged with those from other sets.*

The proof can be found in **Appendix** C.4. Theorem 4, in fact, provides a simplified solution from the perspective of variable merging. However, this solution is subject to conditions and still requires partial knowledge of the underlying causal relationships. Nevertheless, the theorem offers a principled approach to precise causal modeling and serves as a theoretical foundation for it.

## 3.3 EXPERIMENTAL ANALYSIS

### 3.3.1 RWG DATASET

Table 1: Comparison between our proposed RWG dataset and existing benchmark datasets.

| Dataset | Adjustable Elements (↑) | Known Causality | Adjustable Motif | Adjustable Node Feature | Adjustable Edge Connection | Adjustable Assemble Mode | Real-world Grounded Data |
|---|---|---|---|---|---|---|---|
| Citeseer (Giles et al., 1998) | Fixed | ✗ | ✗ | ✗ | ✗ | ✗ | ✓ |
| PROTEINS (Morris et al., 2020) | Fixed | ✗ | ✗ | ✗ | ✗ | ✗ | ✓ |
| Synthetic Graph (Ying et al., 2019) | 5 | ✓ | ✓ | ✗ | ✗ | ✗ | ✗ |
| Spurious-Motif (Wu et al., 2022b) | 6 | ✓ | ✓ | ✗ | ✗ | ✗ | ✗ |
| CRCG (Gao et al., 2024) | 54 | ✓ | ✓ | ✓ | ✓ | ✗ | ✗ |
| RWG | 90 | ✓ | ✓ | ✓ | ✓ | ✓ | ✓ |

To further investigate the research problem while maintaining close ties to real-world scenarios, we introduce the **R**eal-**W**orld knowledge-based synthesized **G**raph (RWG) dataset for empirical analysis. This dataset is

grounded in real-world knowledge and rules, leveraging chemical and citation networks to construct synthetic graph samples for testing graph classification and node classification tasks. Specifically, the RWG dataset generates graph samples that closely resemble real-world chemical molecules by integrating various chemical motifs, connecting modules, and controllable parameters, ensuring clear and modelable internal causal relationships. It also simulates node features from real-world citation networks, constructing graph samples that approximate real-world citation structures, with known internal causal relationships. Table 1 compares the RWG dataset with other related datasets. For additional details, please refer to **Appendix** D.

Next, we conduct experiments using the RWG dataset we have constructed to cross-validate with the previous theoretical analysis.

### 3.3.2 CAUSAL MODELING CAPABILITY

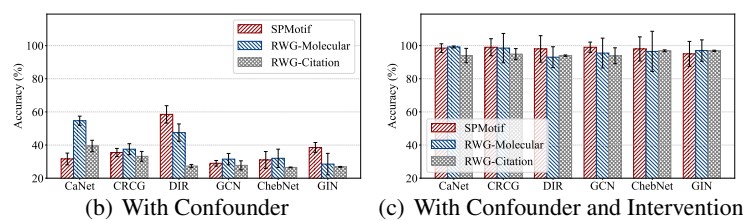

(a) Without Confounder     (b) With Confounder     (c) With Confounder and Intervention

Figure 3: Test accuracy comparison across three different scenarios.

We first analyze the causal modeling ability of multiple GNN baselines when dealing with different datasets and the impact of intervention operations on this ability. Three datasets are used: SPMotif (Wu et al., 2022b), RWG-Molecular, and RWG-Citation. Among them, RWG-Molecular and RWG-Citation are graph classification and node classification datasets based on RWG, while SPMotif is a widely used artificially synthesized graph dataset. The causal relationships in these datasets can be formally modeled and precisely controlled.

Six GNN baselines are used for experimental analysis, including three causal relationship modeling-enhanced GNN baselines: CaNet (Wu et al., 2024), CRCG (Gao et al., 2024), DIR (Wu et al., 2022b), and three general GNN baselines: GCN (Kipf & Welling, 2017), ChebNet (Defferrard et al., 2016), and GIN (Xu et al., 2019). We first make the causal relationships in the dataset explicit, ensuring that there is no interference from confounders, so that the probabilistic associations in the dataset are equivalent to the causal associations. In other words, all the elements we use to construct the dataset are associated with the labels. At the same time, by reducing the problem's difficulty and conducting multiple rounds of training, we make the GNN modeling performance approach 100% test accuracy. Then, we introduce confounders into the dataset and apply interventions to observe the effects. When applying interventions, we follow the approach used by other causal graph representation learning methods (Wu et al., 2022b; Gao et al., 2024), treating the confounder as a whole for the intervention. However, since we have complete knowledge of the internal causal relationships within the data, we can ensure that the division of causal variables complies with Theorem 4. Specifically, we fix the confounder as specific, invariant graph data to eliminate interference and perform the intervention. Please refer to **Appendix** F for baselines and dataset details.

Figure 3 shows that introducing confounders degrades model performance, while interventions significantly improve accuracy, almost fully restore the no-confounder level. This result supports Theorem 4, demonstrating that intervention-based causal inference remains effective under reasonable variable merging. At the same time, it can be observed that there is still some inevitable performance degradation in real-world scenarios. This is related to the inherent limitations of GNN models and the inability of intervention methods applied to graph data to completely eliminate interference.

### 3.3.3 INTERVENTION ANALYSIS

In this section, we analyze intervention effects under varying conditions using the same datasets and baselines as in Section 3.3.2. We simulate errors in variable merging by reassigning parts of $X^{\text{cfd}}$ to $X^{\text{caus}}$, thereby violating Theorem 4. Given that our dataset is artificially synthesized with controllable causal relationships, this operation merely entails treating the confounders as non-intervened components when applying causal interventions. Results are shown in Figure 4. As the violation increases, model performance degrades, indirectly validating Theorem 4. Moreover, SPMotif shows smaller fluctuations than RWG-Molecular and RWG-Citation, due to its simpler structure. This highlights the importance of RWG datasets, which better approximate real-world complexity and yield more reliable experimental outcomes.

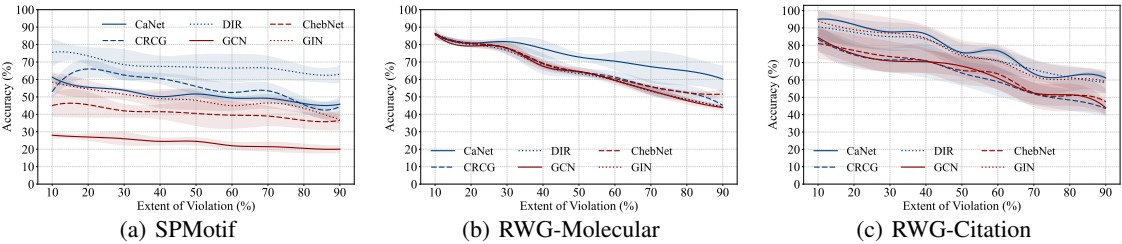

|  (a) SPMotif | (b) RWG-Molecular | (c) RWG-Citation |

Figure 4: Performance of the methods when Theorem 4 is violated to varying degrees. The horizontal axis represents the percentage of data in $X^{\text{cfd}}$ that is erroneously merged into $X^{\text{caus}}$.

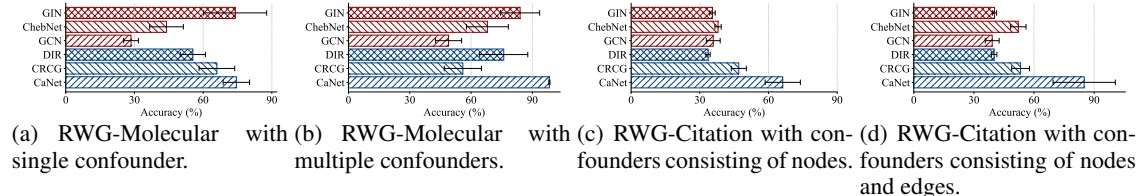

(a) RWG-Molecular with single confounder.    (b) RWG-Molecular with multiple confounders.    (c) RWG-Citation with confounders consisting of nodes.    (d) RWG-Citation with confounders consisting of nodes and edges.

Figure 5: Performance comparison across different scenarios.

We also investigated how different graph elements affect the causal modeling capability of GNNs. Using RWG-Molecular, we compared a single large confounder subgraph with multiple smaller ones; using RWG-Citation, we added confounders consisting only of nodes versus those involving both nodes and edges. Results in Figure 5

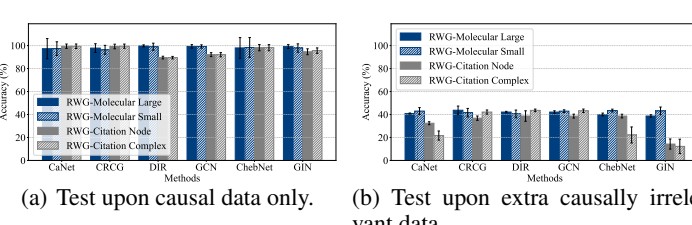

(a) Test upon causal data only.    (b) Test upon extra causally irrelevant data.

Figure 6: Performance of models trained solely with causal data.

show notable performance gaps between RWG-Molecular and RWG-Citation, but only minor differences within the same dataset type. This suggests that merged elements cannot be completely treated as general causal variables, as their effects remain dataset- and scenario-dependent.

Furthermore, we evaluate the generalization of GNNs trained solely on causal relationships using four datasets: RWG-Molecular Large (fewer samples, larger motif), RWG-Molecular Small (more samples, smaller motif), RWG-Citation Node (node-only relations), and RWG-Citation Complex (nodes, edges, and

interactions). Results in Figure 6(a) show good performance on purely causal test data, but adding 70% confounding data (Figure 6(b)) causes a sharp decline. This experiment strongly demonstrates that, even when trained solely on causally associated data, models in complex graph representation learning scenarios remain effective only within the original data distribution. Their performance deteriorates substantially when exposed to extraneous data. Further experimental results can be found in **Appendix** E.2 and E.3, the setting and dataset details can be found in **Appendix** F.

## 4 METHOD

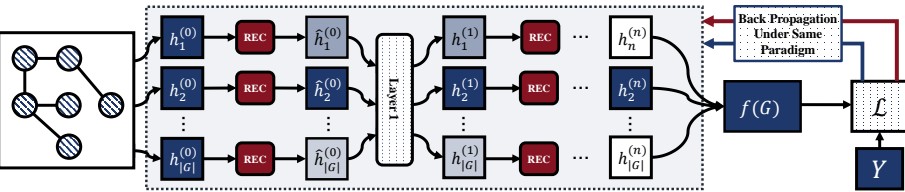

Figure 7: The illustration of how REC works.

Based on the above discussion, within the current graph representation learning framework, achieving strictly accurate causal relationship modeling is nearly impossible. Reviewing our entire analysis, we identify the inherent complexity of graph data as the fundamental obstacle to causal modeling in graph representation learning. Therefore, we consider whether reducing this complexity could enable GNNs to better approximate the true underlying causal model. To begin, we propose the following proposition:

**Proposition 5** *When the GNN $f(\cdot)$ precisely models the causal mechanism, the cross-entropy loss $\mathcal{L}$ between predictions $f(G)$ and labels $Y$ is minimized. Moreover, $\mathcal{L}$ equals the conditional KL divergence between the predictive distribution of $f(\cdot)$ and the background causal model, given each input graph $G$.*

The proof can be found in **Appendix** C.5. Given that $\mathcal{L}$ represents the conditional KL divergence between the trained GNN and the causal model, we argue that reducing data complexity, while keeping $\mathcal{L}$ close to optimal, will make it easier to approximate the background causal model and reduce the likelihood of interference. Moreover, this complexity-reduction approach can be implemented as a plug-and-play module, improving both GNN backbones and causal enhanced graph learning methods.

In light of this, we propose a **R**edundancy **E**limination method for **C**ausal graph representation Learning (REC) to eliminate as many redundant variables as possible in $X^{\text{cfd}}$ and $X^{\text{asoc}}$, thereby simplifying the causal modeling process. The REC extracts the feature $h_v^{(0)}$ of each node $v$ in the graph data $G$, along with the feature $h_v^{(l)}$ after processing through the $l$-th layer, and performs variable removing as follows:

$$\tilde{h}_v^{(0)} = \text{REC}(h_v^{(0)}) = \text{sigmoid}\Big(\gamma + \delta^{(0)}\big(h_v^{(0)}\big)\Big) \cdot h_v^{(0)}, \tag{1}$$

similarly:

$$\tilde{h}_v^{(l)} = \text{REC}(h_v^{(l)}) = \text{sigmoid}\Big(\gamma + \delta^{(l)}\big(h_v^{(l)}\big)\Big) \cdot h_v^{(l)}. \tag{2}$$

Here, $\delta^{(l)}(\cdot)$ is a multilayer perceptron (MLP) with an output dimension of 1, and all $\delta^{(l)}$ within the same layer share the same parameters. sigmoid$(\cdot)$ denotes the sigmoid function, which serves as a masking operator for node features. Depending on the value of $\Big((\gamma + \delta^{(l)}\big(h_v^{(l)}\big)\Big)$, the sigmoid function suppresses certain feature values toward zero, thereby excluding them from the forward propagation process and effectively removing the corresponding variables. $\gamma$ is a value that gradually decreases during the training process. It

is designed to eliminate fewer variables at the beginning, allowing the GNN to first model relationships and then eliminate more variables in later stages. This enables the GNN to remove more redundant variables based on accumulated knowledge. Formally, we have:

$$\gamma = \max\left(\gamma_{\text{init}} \cdot (1 - \epsilon)^t, \ \gamma_{\min}\right), \tag{3}$$

where $t$ denotes the number of current round. $\gamma_{\text{init}}$ and $\gamma_{\min}$ are hyperparameters that set the initial and minimum values, respectively. $\epsilon$, also a hyperparameter, is a small value greater than zero that controls the rate of decrease. Then, the GNN layer that applies REC can be formulated as:

$$h_v^{(l+1)} = \text{Aggregate}\left(\mathbf{W}^{(l)}\text{REC}(h_v^{(l)}), \left\{\mathbf{W}^{(l)}\text{REC}(h_u^{(l)}), \forall u \in \mathcal{N}(v)\right\}\right), \tag{4}$$

where $\mathbf{W}^{(l)}$ denotes the weight matrix for the $l$-th layer, $\mathcal{N}(v)$ denotes the neighboring nodes of node $v$, Aggregate($\cdot$) denotes the aggregation process of GNN. REC can be applied to any GNN encoder in causal graph representation learning methods or GNN backbones to enhance the algorithm's causal modeling capabilities. Parameters within REC are updated along with those of the GNN. Figure 7 offers an illustration of applying REC. To validate its effectiveness, we conducted extensive experiments on multiple datasets. Besides our own proposed datasets, we utilized the artificially generated dataset SPMotif, and real-world datasets CiteSeer (Caragea et al., 2014) and ENZYMES (Rossi & Ahmed, 2015). Furthermore, we merged RWG's link construction paradigm with the SPMotif dataset to construct SPMotif-M, a graph dataset containing more diverse types of graph structural linkages. Simultaneously, we integrated RWG motifs approximating real-world molecular structures with SPMotif to create SPMotif-C.

| Method | RWG-Molecular | Spmotif-M | Spmotif-C | RWG–Citation | CRCG | CiteSeer | ENZYMES |
|---|---|---|---|---|---|---|---|
| CaNet | 52.17 ± 2.02 | 32.40 ± 1.30 | 45.00 ± 1.32 | 59.33 ± 1.04 | 32.77 ± 1.27 | 83.87 ± 0.67 | 17.00 ± 6.78 |
| CaNet+REC | 56.50 ± 2.65 | 34.17 ± 1.35 | 46.83 ± 3.06 | 61.83 ± 0.76 | 36.43 ± 1.17 | 84.90 ± 0.12 | 18.33 ± 6.94 |
| Improvement | +4.33 | +1.77 | +1.83 | +2.50 | +3.66 | +1.03 | +1.33 |
| CRCG | 45.50 ± 3.53 | 36.80 ± 1.89 | 44.50 ± 2.91 | 45.50 ± 6.29 | 29.70 ± 4.62 | 42.78 ± 3.76 | 34.67 ± 7.10 |
| CRCG+REC | 45.50 ± 4.39 | 38.17 ± 3.83 | 50.50 ± 3.05 | 47.50 ± 6.26 | 33.10 ± 5.29 | 44.07 ± 3.30 | 40.33 ± 3.24 |
| Improvement | +0.00 | +1.37 | +6.00 | +2.00 | +3.40 | +1.29 | +5.66 |
| DIR | 49.00 ± 5.26 | 38.67 ± 4.57 | 63.00 ± 6.55 | 52.50 ± 5.67 | 31.80 ± 4.76 | 66.53 ± 1.42 | 42.67 ± 6.01 |
| DIR+REC | 52.00 ± 5.68 | 39.97 ± 3.12 | 67.00 ± 4.79 | 57.50 ± 6.24 | 36.10 ± 4.14 | 67.70 ± 1.74 | 48.00 ± 2.21 |
| Improvement | +3.00 | +1.30 | +4.00 | +5.00 | +4.30 | +1.17 | +5.33 |
| GCN | 40.00 ± 5.56 | 38.60 ± 1.71 | 19.50 ± 2.36 | 43.50 ± 6.96 | 17.22 ± 1.26 | 71.08 ± 0.48 | 24.67 ± 1.94 |
| GCN+REC | 42.35 ± 4.10 | 40.21 ± 0.96 | 26.36 ± 2.76 | 52.29 ± 4.41 | 26.30 ± 1.67 | 71.93 ± 0.33 | 28.33 ± 2.83 |
| Improvement | +2.35 | +1.61 | +6.86 | +8.79 | +9.08 | +0.85 | +3.66 |
| ChebNet | 41.00 ± 4.45 | 38.63 ± 1.61 | 33.50 ± 4.90 | 55.50 ± 7.23 | 33.75 ± 1.89 | 55.39 ± 2.44 | 26.33 ± 3.09 |
| ChebNet+REC | 50.18 ± 6.77 | 40.40 ± 0.98 | 37.95 ± 5.21 | 57.40 ± 6.19 | 36.02 ± 1.91 | 57.27 ± 1.72 | 30.33 ± 1.63 |
| Improvement | +9.18 | +1.77 | +4.45 | +1.90 | +2.27 | +1.88 | +4.00 |
| GIN | 50.50 ± 8.44 | 14.27 ± 4.43 | 36.50 ± 3.43 | 46.50 ± 4.56 | 28.02 ± 0.82 | 52.80 ± 3.53 | 27.00 ± 4.14 |
| GIN+REC | 55.90 ± 1.74 | 38.60 ± 4.60 | 45.00 ± 3.98 | 53.10 ± 1.38 | 33.64 ± 2.39 | 54.57 ± 2.74 | 33.67 ± 1.87 |
| Improvement | +5.40 | +24.33 | +8.50 | +6.60 | +5.62 | +1.77 | +6.67 |

Table 2: Performance comparison of different methods with and without REC enhancement on various datasets. The improvement row shows the absolute performance gain achieved by REC.

Experimental results, as shown in Table 2, demonstrate that our method achieves improvements across all baselines, with significant enhancements in certain scenarios. This not only validates the effectiveness of REC but also provides supporting evidence for Proposition 5. Detailed settings can be found in Appendix F.

## 5 CONCLUSION

This paper approaches causal modeling in graph representation learning from a theoretical perspective, developing a theoretical model that strictly adheres to the fundamental assumptions of causal inference. Building on this foundation, we conduct in-depth analyses combined with experimental cross-validation and further propose an improved enhancement module.

## REPRODUCIBILITY STATEMENT

All of our theoretical results have been rigorously proven, and the corresponding proofs are provided in Appendix C. Additionally, our experiments and methods include data and code for reproducibility. The code for generating datasets is available in the /gen_datasets directory of the supplementary materials. The generated datasets are provided in the /data directory. The code for loading the datasets and training the models is available in the /models directory. For more details and environment setup, please refer to the README.md in the supplementary materials.

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

## A   USAGE OF LARGE LANGUAGE MODEL

In our paper, we used LLMs to assist with polishing the writing, including correcting grammatical errors and making the sentences more consistent with academic English writing conventions.

## B   EXTENDED RELATED WORKS

### B.1   CAUSAL LEARNING

Causal learning is a learning method based on identifying and modeling causal relationships (Pearl, 2000). Unlike traditional correlation learning, causal learning focuses on the causal effects between variables, i.e., how a change in one variable causes a change in another, rather than merely observing the statistical correlation between them. Causal learning aims to extract causal relationships between variables from data and use these relationships for prediction, reasoning, or decision-making. It typically involves methods such as causal graph models (Kocaoglu et al., 2019), causal inference (Pearl & Mackenzie, 2018), and causal discovery (Zheng et al., 2018), to infer causal structures from observational data. In recent years, with the rise of deep learning, research on causal learning has gradually shifted to the field of neural networks, particularly how to incorporate causal inference into the training and inference processes of neural networks.

Causal learning in neural networks is primarily reflected in the integration of causal inference with deep learning models to improve the performance of neural networks. On the one hand, neural networks mainly model data by learning the relationships between variables, but since they cannot directly understand the causal relationships between variables, this limits their performance on more complex problems. Therefore, recent research has attempted to incorporate causal inference into the training process of neural networks in order to enhance model interpretability and generalization ability (Chattopadhyay et al., 2019; Zhang et al., 2020). Zhang et al. (2021) removes dependencies between features by learning weights for training samples, thus allowing deep learning models to avoid spurious correlations and focus more on the true relationships between features and labels. Yao et al. (2024) analyzes and understands the causal relationships between latent variables in the data, identifying more fine-grained representations under the generally milder assumption of partial observability. Hong et al. (2024) introduces causal models to understand and advance Non-transferable learning by modeling content and style as two latent factors, decoupling them and using them as guides to learn non-transferable representations with inherent causal relationships. These methods enhance the model's reasoning ability by introducing causal graph structures or causal analysis mechanisms into neural networks.

Moreover, causal deep generative models are also an important research direction in causal learning within neural networks in recent years (He et al., 2023; Bagi et al., 2023; Cheng et al., 2024). For example, Sauer & Geiger (2021) proposes decomposing the image generation process into independent causal mechanisms and training them without direct supervision. By utilizing appropriate inductive biases, these mechanisms disentangle object shape, object texture, and background, thus enabling the generation of counterfactual images. In the field of reinforcement learning, causal inference has also started to integrate with deep reinforcement learning methods (Zeng et al., 2023; Zhu et al., 2023; Yu et al., 2024). For example, Zhang et al. (2024) adopts a guided updating mechanism to learn a stable causal origin representation. By leveraging this representation, the learned policy demonstrates significant robustness to nonstationarity.

### B.2   CAUSAL RELATIONSHIPS MODELING WITH GNNs

This paper primarily explores how to enhance the causal relationship modeling in Graph Neural Networks (GNNs). Causality is crucial in graph representation learning (Lippe et al., 2023; Sui et al., 2022), as the complexity and variability of graph data, unlike images and text, require stronger causal relationship

modeling capabilities to ensure generalization and robustness. Moreover, several application areas of graph representation learning, including finance (Wang et al., 2022), medicine (Shang et al., 2019), and biology (Zitnik et al., 2018), have significant demands for the causal relationships being modeled.

There are currently many related studies addressing this issue, which can be categorized into two technical approaches: one focusing on modeling causal relationship subgraphs and the other on eliminating the impact of confounding factors. Regarding the first approach, Fan et al. (2024) proposed that spurious correlations exist within subgraph-level units and analyzed the degeneration of GNNs from a causal perspective. Based on this causal analysis, a general causal representation framework was proposed to build stable GNNs. Wu et al. (2022b) introduced a new discovering invariant rationale (DIR) strategy to construct inherently interpretable GNNs and enhance their causal relationship modeling ability. Chen et al. (2022) proposed a new framework called Causal-Inspired Invariant Graph Learning (CIGA) to capture the invariances in graphs, ensuring out-of-distribution (OOD) generalization under various distribution changes.

Regarding the second approach, Fan et al. (2022) proposed a general decoupled GNN framework, learning causal substructures and bias substructures separately. Gao et al. (2024) developed a lightweight optimization module based on the relationship between causal key modeling and confounding factors. Fan et al. (2022) also introduced a general decoupled GNN framework to separately learn causal and bias substructures, ensuring that the final model can debias. Wu et al. (2024) employed a new learning objective inspired by causal inference, which coordinates an environment estimator with an expert mixed GNN predictor. This new method overcomes the confounding biases in training data and promotes the learning of widely adaptable predictive relationships.

## C  THEORETICAL PROOFS

### C.1  PROOF OF PROPOSITION 1

**proposition 1.** *When the variables within graph dataset $\mathcal{G}$ are merged to form a new and smaller variable set $S$, in certain cases, it becomes impossible to construct a causal model based on $S$ while still satisfying the two key prerequisites for applying causal inference methods—namely, the Causal Markov Assumption and the Causal Faithfulness Assumption.*

*Proof.*    To illustrate the proposition, we provide a corresponding counterexample. To ensure the clarity of the proof, we first present the detailed formulations of the Causal Markov Assumption and Causal Faithfulness Assumption.

**Causal Markov Assumption** (Spirtes, 2009) **:** *For a set of variables in which there are no hidden common causes, variables are independent of their non-effects conditional on their immediate causes.*

**Causal Faithfulness Assumption** (Spirtes, 2009)**:** *There are no independencies other than those entailed by the Causal Markov Assumption.*

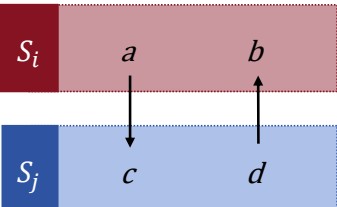

Figure 8: The graphical illustration of the causal relationships between variables $a$, $b$, $c$, and $d$.

For the Causal Markov Assumption, given variables $S_i$ and $S_j$ within $S$, we assume that $\{a, b\} \subset S_i$ and $\{c, d\} \subset S_j$, and that the ground-truth causal relationships among $a$, $b$, $c$, and $d$ are as illustrated in Figure 8. In this case, $S_i$ contains causes of $S_j$, and $S_j$ also contains causes of $S_i$. Suppose we designate $S_i$ as the cause and construct a causal pathway $S_i \rightarrow S_j$. Then, for another variable $S_k \in S$, if $S_k$ also contains causes of $S_j$, and $S_j$ contains causes of $S_k$, while $S_k$ is similarly designated as the cause, it is possible that $S_i$ and $S_k$ are dependent even when conditioned on all their common causes. Moreover, this phenomenon persists regardless of how the causal direction is assigned within the constructed causal model. Thus, the Causal Markov Assumption no longer holds.

For the Causal Faithfulness Assumption, we just need to assume that $a \perp\!\!\!\perp b$; in that case, such an independence would be regarded as arising from factors other than those implied by the Causal Markov Assumption. The proposition is proved.

$\square$

### C.2 PROOF OF THEOREM 2

**Theorem 2** *The SCM in Figure 2 can characterize the general causal relationships between various variables in the graph representation learning scenario. Furthermore, such an SCM satisfies the Causal Markov Assumption and the Causal Faithfulness Assumption.*

*Proof.* To demonstrate the theorem, we follow the PC algorithm (Spirtes & Glymour, 1991), a method used to infer causal relationships from observational data and reconstruct the SCM depicted in Figure 2 from scratch. The entire process can be divided into three steps, which are detailed below.

**Step 1.** As in the current scenario, for any $i \in \{1, 2, ..., m\}$, there does not exist a set $B$ such that the conditional independence $U_i \perp\!\!\!\perp X_i | B$ holds. According to Spirtes & Glymour (1991), we connect each element in $U$ with its corresponding element in $X$. Since it cannot be determined whether there exists a $B$ such that $U_i \perp\!\!\!\perp X_j | B$ holds for $i \neq j, i \in \{1, 2, ..., m\}, j \in \{1, 2, ..., m\}$, we use dashed lines to connect these elements. For the same reasons, we connect all elements in $X$. Additionally, since the following holds:

$$X^{\text{caus}} = \bigcup_{i \in 1,2,...,k} Pa(Y_i), \tag{5}$$

we have:

$$X_i \perp\!\!\!\perp Y \mid \left( \bigcup_{i \in 1,2,...,k} Pa(Y_i) \right), \forall X_i \in X^{\text{cfd}} \cup X^{\text{asoc}}. \tag{6}$$

i.e.:

$$X_i \perp\!\!\!\perp Y \mid X^{\text{caus}}, \forall X_i \in X^{\text{cfd}} \cup X^{\text{asoc}}. \tag{7}$$

Therefore, we only link the elements within $X^{\text{caus}}$ with $Y$ using dashed lines. The result of step 1 is demonstrated in Figure 9(a).

**Step 2.** Since we do not study the values of exogenous variables, we only consider their influence on $X$, hence all edges from $U$ to $X$ as directed downwards. Based on Equation 5, we direct the edges between $X^{\text{caus}}$ and $Y$ towards $Y$. As elements within $X^{\text{cfd}}$ holds none causal path towards $Y$, we direct edges between $X^{\text{asoc}} \cup X^{\text{caus}}$ and $X^{\text{cfd}}$ towards $X^{\text{cfd}}$. The result of step 2 is demonstrated in Figure 9(b).

**Step 3.** The remaining edges cannot be oriented, thus they are represented using bidirectional arrows. The result of step 3 is demonstrated in Figure 9(c).

We can see that the final result obtained, as shown in Figure 9(c), is consistent with Figure 2.

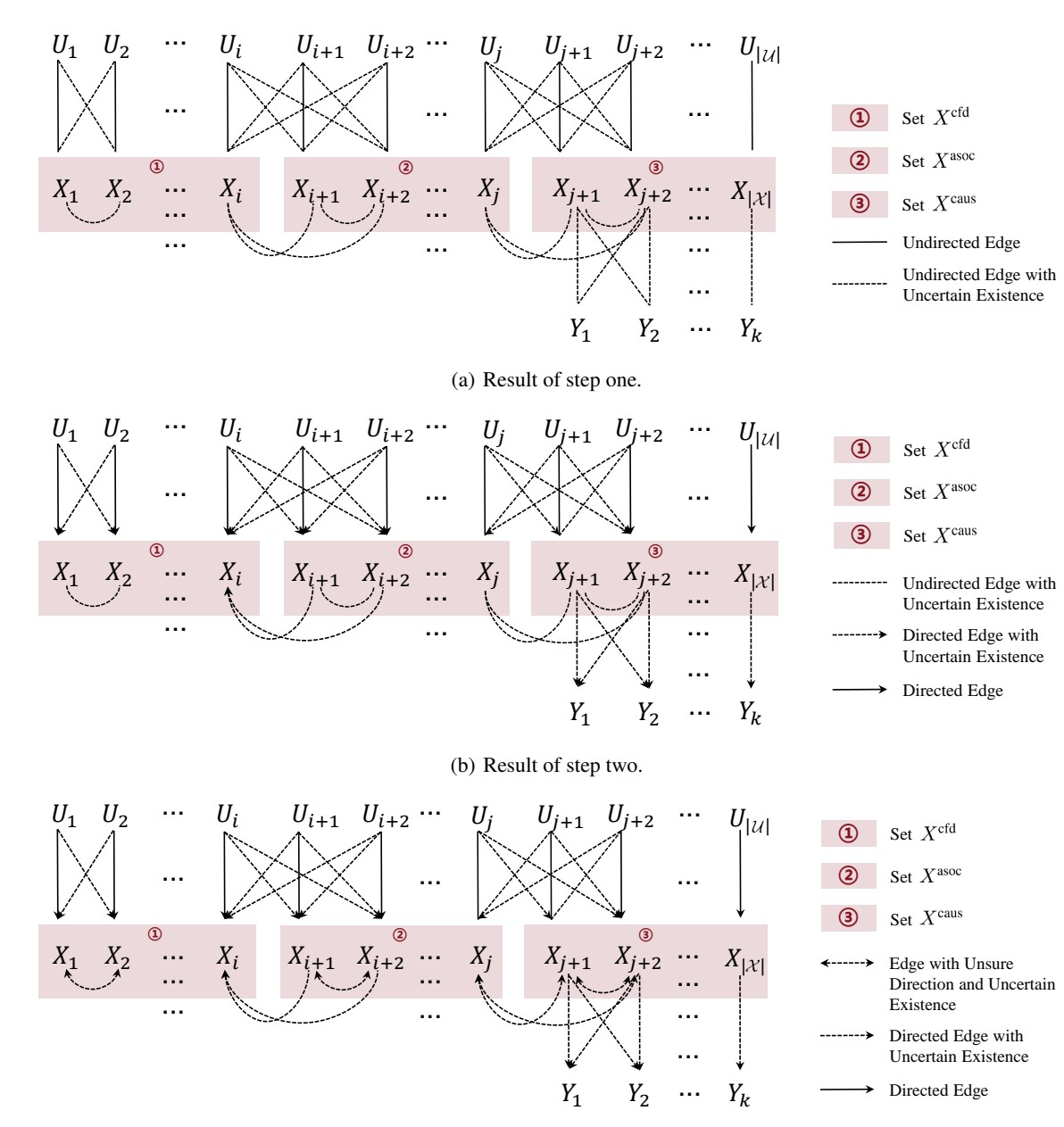

(a) Result of step one.

(b) Result of step two.

(c) Result of step three.

Figure 9: Results of the PC algorithm for SCM reconstruction.

Furthermore, since the variables used in the SCM are the smallest divisible variables in $G$, when there are no common causes between any two variables $a$ and $b$ in the SCM, $Pa(a)$ can block all causal effects from $a$ to $b$. Therefore, the Causal Markov Assumption holds.

At the same time, for any $a \perp\!\!\!\perp b$ in this SCM, $Pa(a)$ can block all causal effects from $a$ to $b$. Thus, the Causal Faithfulness Assumption also holds. The theorem is thereby proven. $\qquad\square$

### C.3 PROOF OF THEOREM 3.

**Theorem 3** *Based on the SCM in Theorem 2, When utilizing GNN to model causal relationship, for atomic interventions, the lower bound of the number of interventions required is* $\min_{\mathcal{M}^{micro}} \left( \left\lceil \dfrac{\frac{1}{\lambda}|\left( \bigcup_{i=1}^{|\mathcal{G}|} G_i \right)|+|Y|-r(\mathcal{M}^{micro})}{2} \right\rceil \right)$*, where $\mathcal{M}^{micro}$ denotes any DAG that is equivalent to the graphical representation of the ground truth causal model, and the vertex set of $\mathcal{M}^{micro} = \left( \bigcup_{i=1}^{|\mathcal{G}|} G_i \right) \cup Y$, $\lambda$ denotes the average times that each variable occurs among each of the samples within dataset $\mathcal{G}$, $r(\cdot)$ calculates the total number of maximal cliques. For non-atomic interventions, the number of interventions required exceeds* $\mathcal{O} \left( \min_k \left( \dfrac{\frac{1}{\lambda}|\left( \bigcup_{i=1}^{|\mathcal{G}|} G_i \right)|+|Y|}{k} log\left( log\left( k \right) \right) \right) \right)$*.*

*Proof.* To conduct the proof, we perform the analysis within the SCM framework shown in Figure 2. Our focus is on the causal relationship between $X$ and $Y$, and thus, we concentrate on the variable relationships between these two sets. As we do not yet have a clear partition of the variables within $X$, the elements within $X$ remain unknown to us. Following the analysis of causal inference and variable definitions presented in Spirtes (2009), it is crucial to ensure that the defined variables satisfy both the Markov assumption and the faithfulness assumption to facilitate accurate causal reasoning.

As noted in Pearl (2009), *starting from the deterministic case, all variables can be explained by microscopic details, ensuring the Markov assumption holds.* Without a clear partition of variables in advance, we need to follow the approach in Pearl (2009) by decomposing all variables to the finest granularity to ensure the Markov assumption holds. Assuming each node in the graph data corresponds to a single-dimensional attribute, every node is treated as an individual variable, ensuring minimal data partitioning. As $\lambda$ denotes the average times that each variable occur among each of the samples within dataset $\mathcal{G}$, the dataset contains $\frac{1}{\lambda}|\left( \bigcup_{i=1}^{|\mathcal{G}|} G_i \right)| + |Y|$ variables.

We denote such variable set as $X^{\mathrm{micro}}$, we have:

$$|X^{\mathrm{micro}}| = \frac{1}{\lambda}|\left( \bigcup_{i=1}^{|\mathcal{G}|} G_i \right)| + |Y| \tag{8}$$

The proposed theorem by Choo et al. (2022) provides the lower bound of the number of atomic interventions required for modeling causal relationships among $\mathcal{M}^{\mathrm{micro}}$, where $\mathcal{M}^{\mathrm{micro}}$ denotes any Markov equivalence class corresponding to $\mathcal{M}^{\mathrm{micro}*}$. $\mathcal{M}^{\mathrm{micro}*}$ is the unknown causal model's graphical representation of all

variables within $X^{\text{micro}}$. Therefore, we have that:

$$
\begin{aligned}
|\mathcal{I}| &\geq \min_{\mathcal{M}^{\text{micro}}} \left( \left\lceil \frac{|X^{\text{micro}}| - r(\mathcal{M}^{\text{micro}})}{2} \right\rceil \right) \\
&\geq \min_{\mathcal{M}^{\text{micro}}} \left( \left\lceil \frac{\frac{1}{\lambda} | \left( \bigcup_{i=1}^{|\mathcal{G}|} G_i \right) | + |Y| - r(\mathcal{M}^{\text{micro}})}{2} \right\rceil \right),
\end{aligned}
\tag{9}
$$

where $\mathcal{I}$ denotes the set of utilized interventions. $r(\mathcal{M}^{\text{micro}})$ denotes the total number of maximal cliques in the chordal chain components of $\mathcal{M}^{\text{micro}}$. Choo et al. (2022) propose that $\mathcal{M}^{\text{micro}}$ is equivalent to $\mathcal{M}^{\text{micro}*}$ if they cannot be distinguished with statistical independency.

If the model we use satisfies the universal approximation theorem (Hornik et al., 1989) and the data in the dataset is sufficient, then we can model $\mathcal{M}^{\text{micro}}$ based on the statistical information from the dataset. Otherwise, more computational effort is needed to solve the structure of $\mathcal{M}^{\text{micro}}$ first. In either case, the lower bound of Equation 9 holds.

For non-atomic interventions, research is still ongoing, and providing an exact calculation of the precise lower bound remains challenging. However, Shanmugam et al. (2015) has provided an approximate estimate of its lower bound, as follows:

$$
|\mathcal{I}| \geq \mathcal{O}\left( \frac{n}{k} log\left( log\left( k \right) \right) \right),
\tag{10}
$$

where $n$ denotes the number of the studied variables, $k$ denotes the number of variables under interventions. Therefore, we acquire the lower of non-atomic interventions as follows:

$$
|\mathcal{I}| \geq \mathcal{O}\left( \frac{\frac{1}{\lambda} | \left( \bigcup_{i=1}^{|\mathcal{G}|} G_i \right) | + |Y|}{k} log\left( log\left( k \right) \right) \right),
\tag{11}
$$

We refine the lower bound for any $k$ as follows:

$$
|\mathcal{I}| \geq \mathcal{O}\left( \min_k \left( \frac{\frac{1}{\lambda} | \left( \bigcup_{i=1}^{|\mathcal{G}|} G_i \right) | + |Y|}{k} log\left( log\left( k \right) \right) \right) \right).
\tag{12}
$$

Based on the above results, we have proved that for atomic interventions, the number of interventions required to fully discern all the causal relationships between the variables in $X$ and $Y$ exceeds $\min_{\mathcal{M}^{\text{micro}}} \left( \left\lceil \frac{\frac{1}{\lambda} | \left( \bigcup_{i=1}^{|\mathcal{G}|} G_i \right) | + |Y| - r(\mathcal{M}^{\text{micro}})}{2} \right\rceil \right)$. For non-atomic interventions, the number of interventions required exceeds $\mathcal{O}\left( \min_k \left( \frac{\frac{1}{\lambda} | \left( \bigcup_{i=1}^{|\mathcal{G}|} G_i \right) | + |Y|}{k} log\left( log\left( k \right) \right) \right) \right)$. Since our goal is to model the causal relationships between $G$ and $Y$, variables that are independent of $Y$ in the graphical data do not need to be analyzed. Therefore, we have:

$$
|\mathcal{I}| \geq \min_{\mathcal{M}^{\text{micro}}} \left( \left\lceil \frac{\frac{1}{\lambda} | \left( \bigcup_{i=1}^{|\mathcal{G}|} G_i \right) | + |Y| - \frac{1}{\sigma} | \left( \bigcup_{i=1}^{|\mathcal{G}|} D_i \right) | - r(\mathcal{M}^{\text{micro}})}{2} \right\rceil \right),
\tag{13}
$$

where $D \in G$ denotes the data that independence with $Y$, $\sigma$ denotes the average times that each variable within $\bigcup_{i=1}^{|\mathcal{G}|} D_i$ occurs. We have:

$$|\mathcal{I}| \geq \min_{\mathcal{M}^{\text{micro}}} \left( \left\lceil \frac{\frac{1}{\lambda}|\left( \bigcup_{i=1}^{|\mathcal{G}|} G_i \right)| + |Y| - r(\mathcal{M}^{\text{micro}})}{2} \right\rceil \right). \quad (14)$$

For non-atomic interventions, we have:

$$|\mathcal{I}| \geq \mathcal{O} \left( \min_k \left( \frac{\frac{1}{\lambda}|\left( \bigcup_{i=1}^{|\mathcal{G}|} G_i \right)| + |Y|}{k} log\left( log\left( k \right) \right) \right) \right). \quad (15)$$

The theorem is proved. □

### C.4   PROOF OF THEOREM 4

**Theorem 4** *Assume there exists a GNN model that satisfies the infinite approximation theorem ([Cybenko](), [1989]), and that interventions are applied to ensure the GNN models the causal relationships between the graph variables and the labels. In this case, when applying causal inference in graph representation learning, it is possible to merge some variables from the original set $X$ to form a new set $S$, where $|S| < |X|$, while ensuring that the causal relationships between the graph data and the labels are accurately modeled. However, the following conditions must be met:*

*(1) Variable $s$ in $S$ that satisfy $s \in Pa(Y)$ cannot simultaneously contain both the parent and child nodes of another variable $v \in X$.*

*(2) Variables within $X^{caus}$ cannot be merged with those of other sets.*

*Proof.*   To prove the theorem, it suffices to demonstrate that the two conditions (1) and (2) proposed in the theorem are both necessary and sufficient for transforming $X$ into $S$, while ensuring the accurate modeling of causal relationships.

We first prove sufficiency. Based on Theorem 2, since conditions (1) and (2) hold, we have the following:

$$P\big(Y \mid do(X \setminus X^{\text{caus}} = x)\big) = P\big(Y \mid do(S \setminus S^{\text{caus}} = x)\big), \quad (16)$$

where $x$ is a specific value of $X \setminus X^{\text{caus}}$, and $S^{\text{caus}}$ denotes the set of variables within $S$ generated by merging $X^{\text{caus}}$. Here, $do(\cdot)$ represents the intervention operation.

Equation 16 demonstrates that conducting the same intervention on $X \setminus X^{\text{caus}}$ and $S \setminus S^{\text{caus}}$ yields identical outcomes. This implies that, with an appropriate set of interventions, even if the variables are merged into set $S$, it remains possible to apply interventions that isolate and sever the influence of variables not involved in the causal component.

We can also conclude the following:

$$P\big(Y \mid do(X^{\text{caus}} = x)\big) = P\big(Y \mid do(S^{\text{caus}} = x)\big). \quad (17)$$

Furthermore, for each $S_i^{\text{caus}} \in S^{\text{caus}}$, we have:

$$P\big(Y \mid do(O_i^{\text{caus}} = s_i)\big) = P\big(Y \mid do(S_i^{\text{caus}} = s_i)\big), \quad (18)$$

and

$$P\big(Y \mid do(X^{\text{caus}} \setminus O_i^{\text{caus}} = s_i)\big) = P\big(Y \mid do(S^{\text{caus}} \setminus S_i^{\text{caus}} = s_i)\big), \quad (19)$$

where $O_i^{\text{caus}}$ is the subset of elements within $X$ that, when merged, create $S_i$.

Based on condition 2 of the proposed theorem, we can model the causal relationship between each $S_i^{\text{caus}}$ and $Y$ via the intervention operation. Therefore, we can apply a suitable GNN as described in the theorem to model the causal relationship between $S^{\text{caus}}$ and $Y$, i.e., the causal relationship between $G$ and $Y$. Thus, sufficiency is proven.

Next, we prove the necessity. First, consider the case where condition (1) does not hold. In this situation, it is possible for two variables, $s$ and $v$, to exist such that $s$ is both a parent and a child of $v$ within the background SCM. This creates a confounding arc (Pearl, 2009) and cannot be removed via intervention. As a result, the causal relationship becomes unpresentable. Thus, condition (1) must hold.

Now, consider the case where condition (2) is violated. In this case, a variable $S_j^{\text{caus}}$ may include components unrelated to the outcome $Y$. When examining $S_j^{\text{caus}}$, the confounding effects cannot be eliminated through intervention, rendering the causal relationship unfeasible. Therefore, condition 2 must also be satisfied. The necessity is proved. Therefore, the theorem is proved. $\square$

## C.5 Proof of Proposition 5

**Proposition 5** *When the GNN $f(\cdot)$ precisely models the causal mechanism, the cross-entropy loss $\mathcal{L}$ between predictions $f(G)$ and labels $Y$ is minimized. Moreover, $\mathcal{L}$ equals the conditional KL divergence between the predictive distribution of $f(\cdot)$ and the background causal model, given each input graph $G$.*

*Proof.* The proof of Proposition 5 is straightforward. The key observation is that the ground-truth labels $Y$ are, under all circumstances, the same as output of the background causal model; the remaining steps follow from standard properties of the cross-entropy loss. For completeness and rigor, we still provide a detailed proof below.

We begin by proving the first conclusion of the proposition. According to the definition of $\mathcal{L}$, we have:

$$\mathcal{L} = \frac{1}{n} \sum_{i=1}^{n} log \frac{1}{\tau(f(G_i))}. \tag{20}$$

As $\tau(\cdot)$ extracts the output probability of the ground truth labels, we have $\tau(f(G_i)) = 1$ when the background causal structure is precisely modeled. We denote $\mathcal{L}^*$ as the value of $\mathcal{L}$ under the former condition. Based on Equation 20, the $\mathcal{L}^*$ can be represented as:

$$\mathcal{L}^* = \frac{1}{n} \sum_{i=1}^{n} log \frac{1}{1} = 0. \tag{21}$$

As $\tau(f(G_i)) \leq 1$, thus $log \frac{1}{\tau(f(G_i))} \geq 0$, and $\mathcal{L} = \frac{1}{n} \sum_{i=1}^{n} log \frac{1}{\tau(f(G_i))} \geq 0$. Therefore, $\mathcal{L}^*$ reaches the minimal value. The first conclusion of the proposition is proved.

Next, we proof the second conclusion. The Conditional KL divergence between the output of $f(\cdot)$ and the ground truth label given different inputs can be formulated as:

$$D_{\text{KL}}(p(Y|G)||q(Y|G)) = \sum_{i=1}^{n} p(G_i) \sum_{Y} p(Y|G_i) log \frac{p(Y|G_i)}{q(Y|G_i)}$$

$$= \sum_{i=1}^{n} \frac{1}{n} \sum_{Y} p(Y|G_i) log \frac{p(Y|G_i)}{q(Y|G_i)}. \tag{22}$$

As $p(Y|G_i) = 0$ if $Y$ is not the ground truth label, therefore:

$$D_{\text{KL}}(p(Y|G)||q(Y|G)) = \frac{1}{n} \sum_{i=1}^{n} 1 \cdot log \frac{1}{q(Y^*|G_i)} = \frac{1}{n} \sum_{i=1}^{n} log \frac{1}{\tau(f(G_i))}, \tag{23}$$

where $Y^*$ denotes the ground truth label. The proposition is proved. $\square$

## D  RWG DATASET

### D.1  DATA CONSTRUCTED BASED ON REAL-WORLD CHEMICAL INFORMATION

We constructed the RWG graph classification data based on extensive real-world chemical knowledge, specifically focusing on molecular structures commonly encountered in the field of chemistry. The construction process began by collecting a total of 26 well-known molecular graph motifs, which serve as the fundamental building blocks for the graphs we aim to analyze. These motifs represent common substructures found in a wide variety of molecules, and they play a critical role in capturing the structural diversity that exists within molecular graphs. Table 3 provides a detailed list and description of these motifs.

Each motif is defined by its own unique arrangement of atoms and bonds, and these motifs can also undergo slight variations. The variations are achieved by adding or removing edges between atoms, which allows us to generate a range of related but distinct molecular structures. This flexibility in modifying the motifs ensures that the graph models are not only diverse but also closely aligned with the variability found in real chemical data.

In addition to these molecular motifs, we also constructed 15 connector modules that are based on common chemical molecular architectures. These modules include well-established structural elements, such as ring structures, chain structures, and various hybrid forms that combine these basic components. These connector modules facilitate the composition of the aforementioned molecular graph motifs into larger, more complex molecular graphs, enabling the representation of a broad spectrum of chemical compounds.

Each connector module is implemented through a corresponding function that allows for customization in terms of size and branching. The `size` parameter enables the adjustment of the module's scale, making it possible to control the overall size of the connected structure. The `branch` parameter, on the other hand, allows for modification of the number of branches that extend from the core structure, providing further flexibility in defining how the motifs are interconnected. By adjusting these parameters, we can create a wide variety of complex molecular graphs that reflect the structural diversity of real-world chemical networks.

### D.2  DATA CONSTRUCTED BASED ON REAL-WORLD CITATION NETWORK INFORMATION

We constructed a citation network based on real-world citation network data, consisting of a total of 25 citation relationships. These relationships were carefully selected to represent a diverse range of connections within the network, capturing the complexity of academic citation patterns across various fields. The citation relationships reflect the way in which research papers influence one another through references, and the resulting network serves as a model for understanding the dynamics of knowledge dissemination and academic collaboration. Table 5 presents a detailed overview of these citation relationships, showcasing the various connections between the papers and their corresponding citation patterns.

In parallel with the citation relationships, we developed a total of 24 node feature generation methods, each based on different statistical distributions and mathematical sequences. These methods were designed to generate meaningful node features that reflect both the structural and contextual aspects of the citation network. The details of these feature generation methods, including the specific distributions and sequences used, are presented in Table 6. The distributions encompass a wide range of statistical models, such as normal, uniform, exponential, and lognormal distributions, among others, while the sequences include arithmetic, geometric, Fibonacci, and prime number sequences, providing a rich variety of feature generation options.

To generate the node features for the dataset, we set parameters for each of the distributions and sequences based on the specific characteristics of the citation network. For instance, the parameters were chosen to align with the real-world distribution of citation frequencies, as well as the structural properties of the ci-

tation relationships, such as the number of citations a paper typically receives and how those citations are distributed across the network. Once the node features were generated, we incorporated the citation relationships to guide the construction of the entire graph, ensuring that the generated features were appropriately aligned with the network's structure and that the graph accurately represented the interplay between the different academic papers. Once completed, we constructed a dataset with clear internal causal relationships, closely aligned with real-world scenarios, and it can serve as a foundation for adding confounders and other elements required for experiments.

### D.3 CAUSAL AND CONFOUNDING DATA GENERATION

For our data generation, we establish causality through precise programmatic control over the dataset's construction, enabling us to explicitly define causal factors and introduce confounders. Specifically, we create a causal relationship by making a graph's label directly dependent on designated graph elements, while a confounder is created by introducing a spurious correlation that exists exclusively within the training set and is broken in the validation and test sets. The primary graph elements we manipulate for these purposes are our constructed motifs, node features, and relational edges.

For instance, in our chemical graph dataset built from 26 molecular motifs, a graph's label can be causally determined by the presence of a "benzene ring" motif. To introduce a confounder, a different motif, like a "chain structure," could be made highly correlated with the label in the training data, a correlation that is removed in the test set to make it a misleading shortcut.

Similarly, using our 24-node feature generation methods for the citation network, we can establish a causal link where the label is determined by a statistical property, such as the average value of a feature generated from a Fibonacci sequence. As a confounder, a separate feature from a uniform statistical distribution could be artificially correlated with the label only during training, a pattern that would not hold during evaluation.

Finally, relational edges, representing 25 types of citation relationships, are also precisely controlled. A causal factor could be the existence of a double bond within the presence of a "self-citation" relationship in the citation network. A confounding relationship could be introduced where a specific type of citation link is frequently associated with a positive label in the training data, but this pattern is randomized in the test set to ensure it's a non-causal artifact.

## E EXTRA EXPERIMENTS

In this section, we present additional experimental results to facilitate a more thorough and in-depth analysis. These results provide further insights into the behavior and performance studied methods, enabling a better understanding of the underlying properties.

### E.1 EXPERIMENTAL RESULTS ACROSS DIFFERENT LEVELS OF CONFOUNDER INFLUENCE.

Based on the provided results in Figure 10, the experimental results of different models under varying confounder bias proportions can be analyzed. Causal enhancement methods such as CaNet, CRCG, and DIR generally outperform standard GNN frameworks (GCN, ChebNet, GIN) as the bias increases, indicating that these methods exhibit stronger robustness when faced with biases or noise in the data.

As the confounder bias increases from 10% to 80%, the performance of models like GCN, ChebNet, and GIN declines significantly, suggesting that these models struggle more to recognize underlying patterns when strong biases are present, making them more susceptible to the influence of confounders. Among the causal enhancement methods, DIR shows a slight advantage at higher bias levels (e.g., 50%, 60% and above), indicating that DIR may be more effective in handling and mitigating the impact of confounders

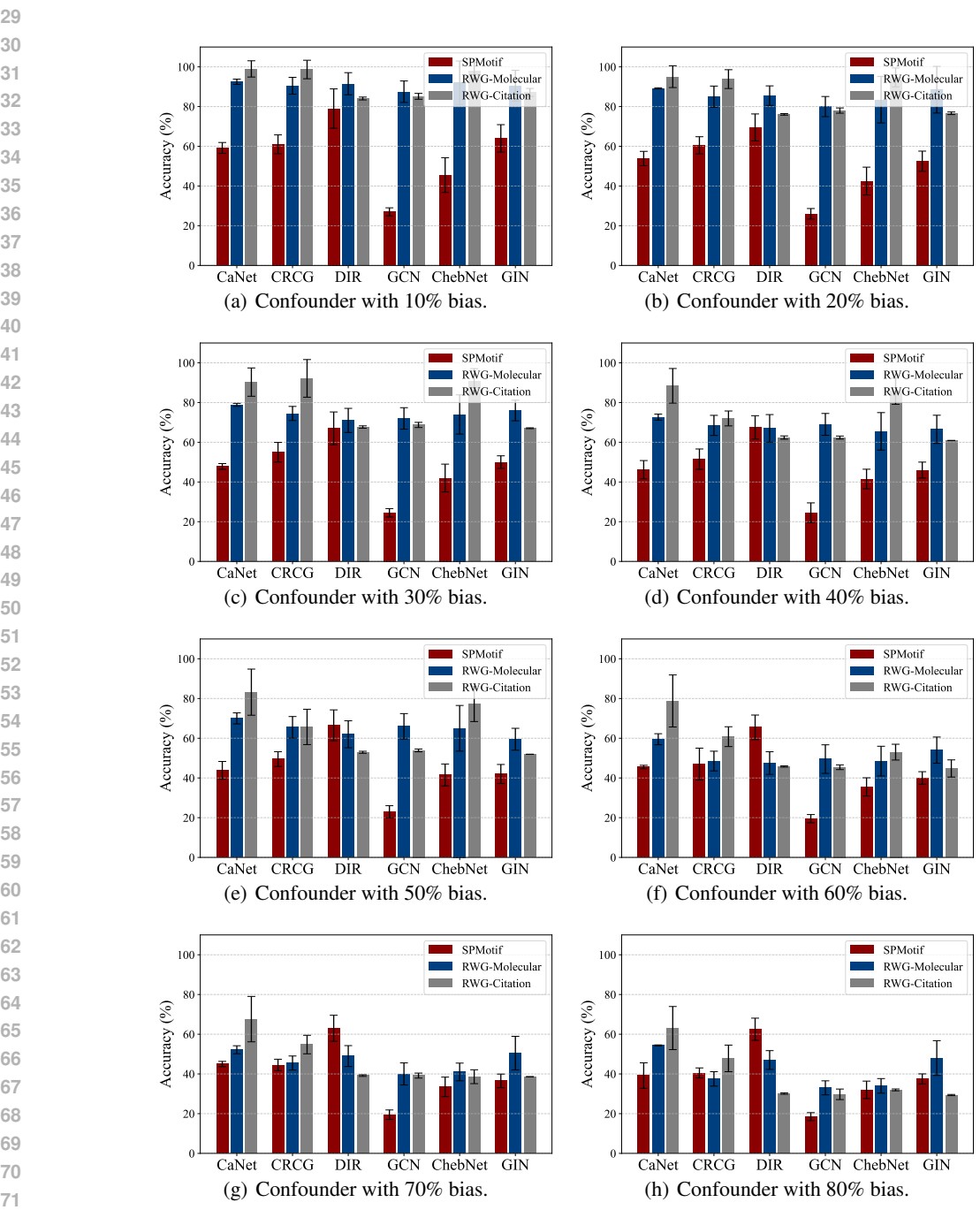

Figure 10: Test Accuracy comparison with different bias.

| Molecule | Molecular Formula | Node Count | Edge Count |
|---|---|---|---|
| Acetic Acid | $C_2H_4O_2$ | 3 | 2 |
| Adrenaline | $C_9H_13NO_3$ | 5 | 6 |
| Ammonia | $NH_3$ | 2 | 3 |
| Anthracene | $C_14H_10$ | 24 | 12 |
| Benzene Ring | $C_6H_6$ | 6 | 6 |
| Benzoic Acid | $C_7H_6O_2$ | 9 | 8 |
| Ethane | $C_2H_6$ | 2 | 1 |
| Ethanol | $C_2H_6O$ | 3 | 2 |
| Fullerenes | $C_60$ | 60 | 90 |
| Glucose | $C_6H_12O_6$ | 24 | 12 |
| Hexamethylbenzene | $C_9H_12$ | 21 | 15 |
| Hydrated Sulfuric Acid | $H_2SO_4 \cdot H_2O$ | 4 | 5 |
| Imidazole | $C_3H_4N_2$ | 9 | 6 |
| Indole | $C_8H_7N$ | 15 | 9 |
| Methane | $CH_4$ | 1 | 1 |
| Methyl Anthranilate | $C_8H_9NO_2$ | 18 | 12 |
| Nitrobenzene | $C_6H_5NO_2$ | 9 | 9 |
| Nitrophenol | $C_6H_5NO_3$ | 10 | 10 |
| Porphyrin | $C_20H_12N_4$ | 24 | 23 |
| Pyridine | $C_5H_5N$ | 6 | 5 |
| Pyrimidine | $C_4H_4N_2$ | 8 | 5 |
| Pyrrole | $C_4H_5N$ | 6 | 5 |
| Simplified Dopamine | $C_8H_11NO_2$ | 11 | 11 |
| Thiazole | $C_3H_3N_S$ | 7 | 5 |
| Thioether | $C_4H_8S$ | 12 | 7 |
| Vitamin C | $C_6H_8O_6$ | 20 | 10 |

Table 3: Fundamental molecular motifs.

compared to other causal methods. Overall, as the confounder bias increases, the performance of all models declines, but the rate of decline varies across different models.

Causal enhancement models exhibit relatively stable performance, especially CaNet and CRCG, which maintain higher accuracy across various bias levels. However, when the confounder bias reaches 90%, even these models experience a significant drop in performance. The accuracy in the "Paper" column (representing some baseline or paper-defined method) consistently remains low, suggesting that traditional methods without causal modeling perform worse when bias is introduced.

In conclusion, causal enhancement methods like CaNet, CRCG, and DIR are more robust to the influence of confounders compared to standard GNN models such as GCN, ChebNet, and GIN, with their advantage being more pronounced at higher bias levels. However, even these causal enhancement methods experience performance degradation under strong biases, indicating that while causal modeling helps mitigate the impact of confounders, it is not immune to them.

## E.2 TRAINING PROCESS ANALYSIS

We also analyzed the training performance of different methods under various scenarios. The results are shown in Figure 11. We can observe significant differences in the training performance of different methods across different datasets. In the molecular dataset with a single confounder (Figure a), the performance of the CaNet model is clearly superior to other methods. As the number of training epochs increases, its validation accuracy continuously rises, ultimately approaching 100%. Other methods, such as GCN, ChebNet, and GIN, show relatively flat performance, with validation accuracy fluctuating around 50%, and they fail to improve significantly.

In the molecular dataset with multiple confounders (Figure b), CaNet still performs excellently, with its validation accuracy surpassing 80% and steadily increasing. Similar to Figure a, the performance of other

| Motif Name | Construction Method | Functionality Description |
|---|---|---|
| Star Motif | Central node connected to all others | Generates a star structure with one center node connected to all peripheral nodes. |
| Path Motif | Nodes connected in sequence | Constructs a linear path where each node connects to the next in sequence. |
| Fan Motif | Central node with multiple branch nodes | Creates a fan-like shape with a central hub and several outward branches, possibly interconnected. |
| Cusped Polygon Motif | Polygon with potential branches | Builds a polygon structure with pointed (cusped) corners and optional branching substructures. |
| Random Bipartite Motif | Bipartite graph with random connections | Generates a bipartite graph where two partitions are randomly interconnected. |
| Tree Motif | Hierarchical branching structure | Constructs a tree graph where each node may connect to multiple child nodes. |
| Trident Motif | Central node with two side branches | Creates a trident-shaped structure, with a central node connected to two others, repeated for multiple tridents. |
| Conical Connection Motif | Backbone and branches in conical form | Forms a cone-like motif where a backbone and branches are merged into a sandglass-shaped structure. |
| Chain Bypass Motif | Chain with branching bypasses | Builds a chain structure with additional side branches that bypass parts of the chain. |
| Partial Polygon Motif | Incomplete polygon with extensions | Forms a partial polygon with potential branch-based extensions. |
| Complete Graph Motif | All nodes interconnected | Constructs a complete graph where every node is connected to every other node. |
| Grid/Net Motif | Nodes arranged in a grid | Creates a net or grid shape where nodes are placed in a matrix and connected to adjacent nodes. |
| Cycle Motif | Nodes forming a ring | Forms a cycle where each node links to the next in a loop. |
| Dual Ring Motif | Two connected ring structures | Builds two separate ring structures that may be interconnected. |
| Triangle Motif | Nodes forming triangles | Creates triangle-based motifs where nodes are connected in three-node cycles. |

Table 4: Connector modules.

| Link Rule | Description |
|---|---|
| Random Citation Generation | Each paper randomly cites a set of papers; the number of citations follows a Poisson distribution. |
| Citation by High Citation Count | Each paper cites papers with higher citation counts to increase connectivity among highly cited papers. |
| Co-Author Based Citation | Papers by authors who have collaborated with the current paper's authors are preferentially cited. |
| Propagation-Based Citation | Citation links are simulated using an information diffusion model (e.g., Independent Cascade). |
| Topic Similarity-Based Citation | Papers with high topic similarity (e.g., via cosine similarity on keywords/abstracts) are cited. |
| Temporal Citation | Older papers are preferentially cited to simulate time-evolving citation behavior. |
| Author Influence-Based Citation | Papers by more influential authors are more likely to be cited. |
| Co-Citation Frequency-Based Citation | Papers that are frequently co-cited with the current paper are selected as citation targets. |
| Citation Density-Based Citation | Papers with higher citation density (degree) are more likely to be cited. |
| Network Topology-Based Citation | Papers cite one of their neighbor nodes; if no neighbors exist, no citation is made. |
| Author Expertise-Based Citation | Papers from authors in the same or related research domains are preferred as citations. |
| Citation Centrality-Based Citation | Papers with higher centrality (e.g., degree, betweenness) in the citation graph are favored. |
| Geographic Proximity-Based Citation | Authors are more likely to cite papers from geographically proximate researchers. |
| Research Team Size-Based Citation | Papers from authors with similarly sized research teams are favored. |
| Citation Credibility-Based Citation | Papers with higher credibility (e.g., journal impact, author reputation) are more likely to be cited. |
| Academic Lineage-Based Citation | Papers authored by academic mentors or descendants are favored. |
| Citation Structure-Based Citation | Triangular citation patterns (e.g., A→B→C→A) are promoted to reflect structural motifs. |
| Citation Distance-Based Citation | Papers with fewer intermediate citation steps (shorter path length) are more likely to be cited. |
| Knowledge Flow-Based Citation | Knowledge flows from frontier to traditional areas guide citation directionality. |
| Citation Chain Length-Based Citation | Longer citation chains increase the likelihood of being cited. |
| Diversity-Based Citation | Papers with more diverse citation sources (across fields or topics) are more likely to be cited. |
| Reference Count-Based Citation | Papers with more references may appear more informative and are thus more likely to be cited. |
| Research Object-Based Citation | Papers focusing on attractive or high-interest research objects are more likely to be cited. |
| Venue Reputation-Based Citation | Papers published in high-impact journals/conferences are preferentially cited. |
| Open Access-Based Citation | Open access papers are more accessible and thus more likely to be cited. |

Table 5: Citation Link Rules in Citation Network Construction

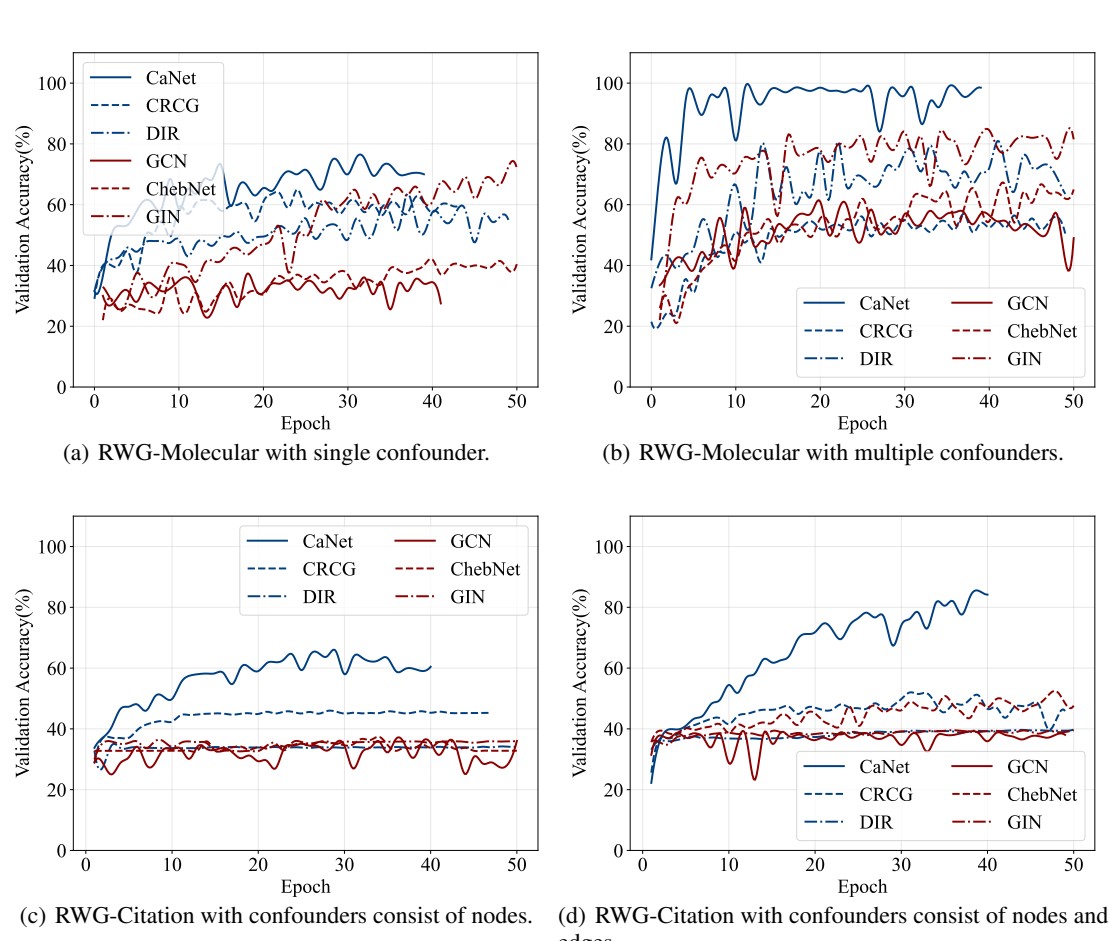

(a) RWG-Molecular with single confounder.

(b) RWG-Molecular with multiple confounders.

(c) RWG-Citation with confounders consist of nodes.

(d) RWG-Citation with confounders consist of nodes and edges.

Figure 11: Validation accuracy upon training procedure.

Table 6: Node Feature Generation Methods based on Statistical Distributions and Mathematical Sequences

| Type | Description |
| --- | --- |
| Normal Distribution | Generates node features based on a normal distribution with specified mean and standard deviation. |
| Uniform Distribution | Generates node features based on a uniform distribution over a specified range. |
| Exponential Distribution | Generates node features based on an exponential distribution. |
| Lognormal Distribution | Generates node features based on a lognormal distribution. |
| Gamma Distribution | Generates node features based on a gamma distribution. |
| Beta Distribution | Generates node features based on a beta distribution. |
| Weibull Distribution | Generates node features based on a Weibull distribution. |
| Laplace Distribution | Generates node features based on a Laplace distribution. |
| Logistic Distribution | Generates node features based on a logistic distribution. |
| Rayleigh Distribution | Generates node features based on a Rayleigh distribution. |
| Pareto Distribution | Generates node features based on a Pareto distribution. |
| Cauchy Distribution | Generates node features based on a Cauchy distribution. |
| Negative Binomial Distribution | Generates node features based on a negative binomial distribution. |
| Gumbel Distribution | Generates node features based on a Gumbel distribution. |
| Gompertz Distribution | Generates node features based on a Gompertz distribution. |
| Arithmetic Sequence | Generates node features based on an arithmetic sequence with a specified step size. |
| Geometric Sequence | Generates node features based on a geometric sequence. |
| Fibonacci Sequence | Generates node features based on the Fibonacci sequence. |
| Square Sequence | Generates node features based on a sequence of square numbers. |
| Cube Sequence | Generates node features based on a sequence of cube numbers. |
| Prime Sequence | Generates node features based on a sequence of prime numbers. |
| Triangular Sequence | Generates node features based on a triangular number sequence. |
| Rectangular Sequence | Generates node features based on a rectangular number sequence. |
| Binomial Coefficient Sequence | Generates node features based on a binomial coefficient sequence. |
| Hamiltonian Sequence | Generates node features based on a Hamiltonian sequence. |

methods is closer to each other, with DIR and GIN showing poorer results, failing to significantly improve the model accuracy.

For the citation dataset, where confounders consist of nodes (Figure c), CaNet's performance remains the most outstanding, with validation accuracy maintained at a high level, and the training process is relatively stable. In Figure d (where confounders consist of both nodes and edges), CaNet still achieves good results, with validation accuracy showing a steady upward trend. Other methods, such as GCN and ChebNet, show slightly worse performance, with validation accuracy fluctuating significantly.

In summary, causal graph representation learning demonstrates an advantage over general methods throughout the training process. Additionally, we observe that some methods may experience performance degradation as training progresses when confounder interference is present.

### E.3 EXPERIMENTAL RESULTS ACROSS DIFFERENT CONFOUNDER TYPES.

We also analyzed the effects of different types of graph elements acting as confounders within Figure 12, 13 and 14. The results are shown in the figures. From the overall trend, it can be observed that the test accuracy fluctuates across different methods as the type of graph element changes.

When dealing with different graph structures, such as "Star," "Path," and "Fan," it is evident that the accuracy of the models varies depending on the confounder type. For example, in certain graph element scenarios, the accuracy fluctuates to varying degrees, while in others, it gradually stabilizes as training progresses, indicating that these methods exhibit different adaptability to confounders.

For molecular structure datasets, such as "Benzene Ring," "Methane," and "Ethane," the impact of different confounder types on model performance is also noticeable. In some structures, the interference of con-

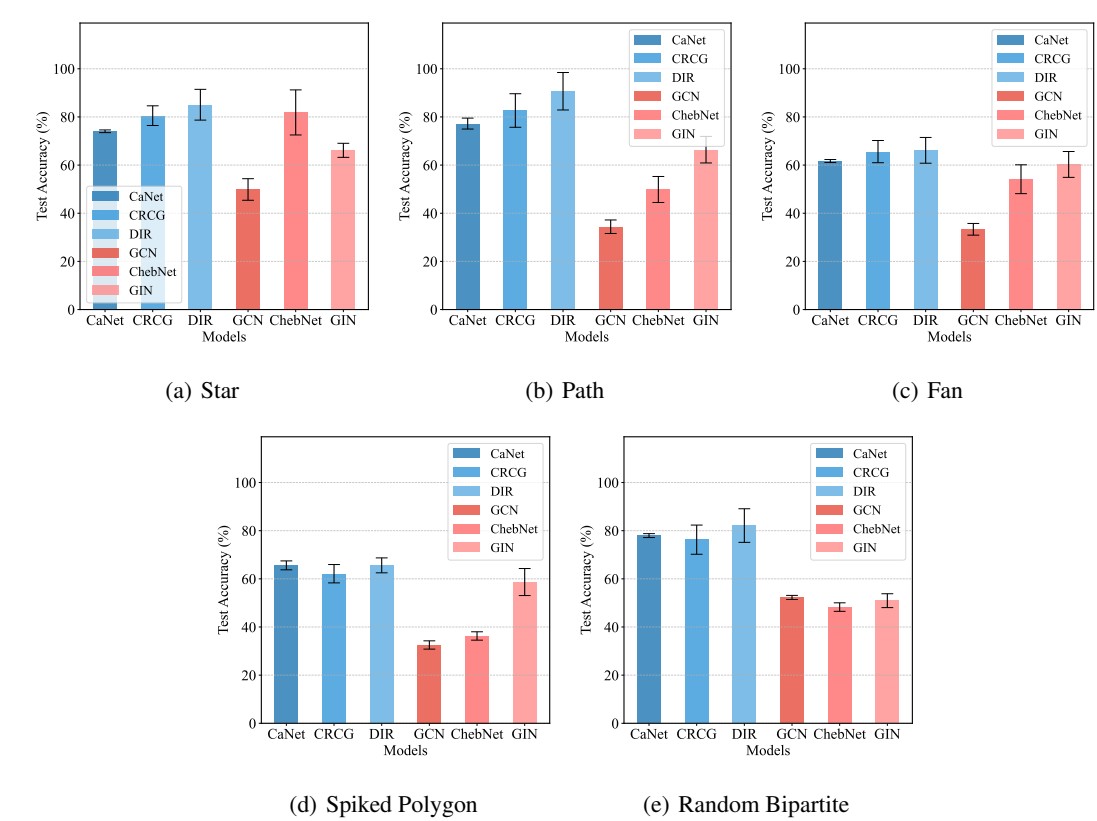

Figure 12: Motif Confounders

founders appears to complicate the training process and affects the final test accuracy, while in other cases, the confounder's interference does not result in a significant accuracy drop.

In the citation dataset, such as "Basic Element," "Citation Element," and "Topic Element," the test accuracy shows more complex trends as the confounder type changes. In certain graph element scenarios, the models exhibit higher volatility when processing specific elements, indicating greater sensitivity to confounders in these contexts.

In conclusion, as the confounder type changes, the test accuracy of the models is influenced to varying degrees, and different types of graph elements exhibit different impact patterns. This suggests that the performance of methods in handling confounders is closely related to the type and complexity of the confounder, as well as the specific structure of the data.

## F  DETAILS OF EXPERIMENTS

### F.1  REC SETTINGS

This module's key hyperparameters include: $\lambda_{\text{init}} = 1.0$, which controls the initial filtering strength; $\epsilon = 0.01$, which implements progressive decay during the training process; $\lambda_{\text{min}} = 0.2$.

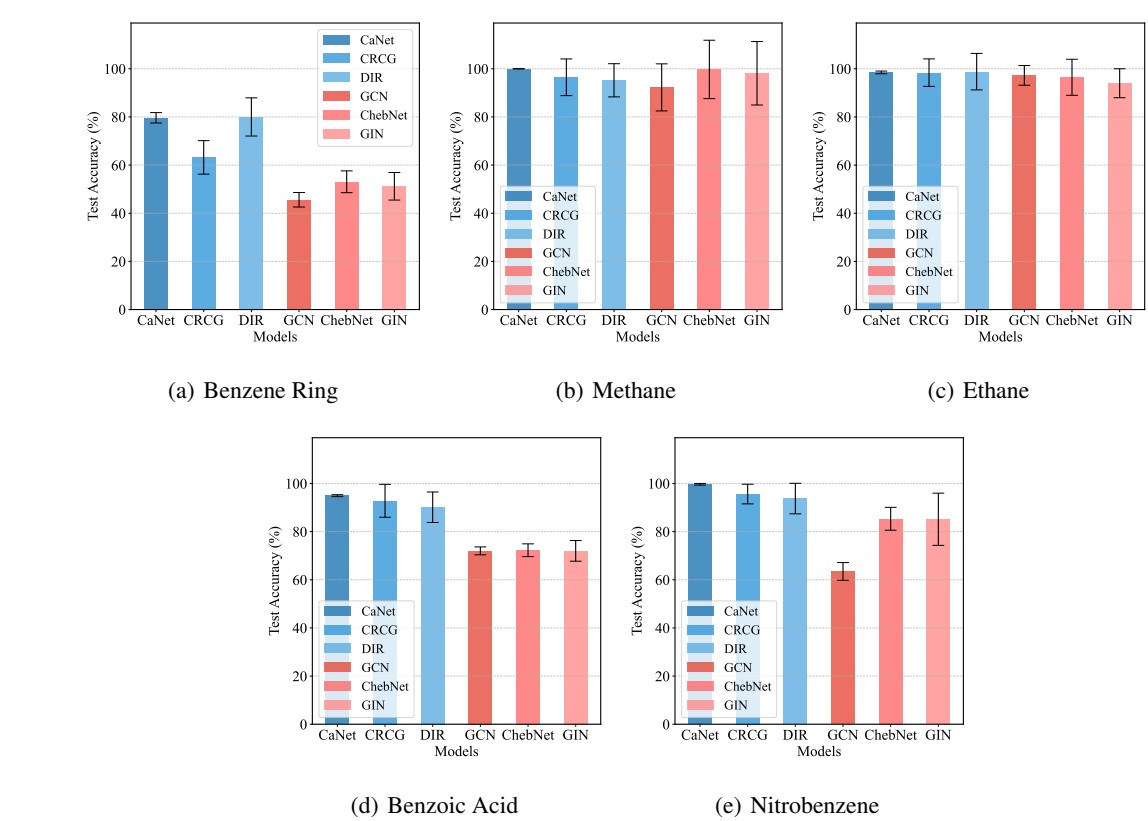

(a) Benzene Ring      (b) Methane      (c) Ethane

(d) Benzoic Acid      (e) Nitrobenzene

Figure 13: Molecular Structure Confounders

## F.2 BASELINES AND SETTINGS

**GCN.** We adopt a two-layer Graph Convolutional Network (GCN) architecture for representation learning. The hidden dimension is set to 64, and each layer performs neighborhood aggregation based on the graph structure, trained using a learning rate of 0.01, a weight decay of $5 \times 10^{-4}$, a batch size of 32, and for 50 training epochs

**GIN.** This baseline is a graph neural network architecture designed to achieve strong expressive power in distinguishing graph structures. It is based on a message-passing mechanism, where node representations are iteratively updated through neighborhood aggregation. In our configuration, GIN employs a hidden dimension of 64, with two network layers. The model is also trained using the setting as GCN.

**ChebNet.** This baseline performs graph convolution through Chebyshev polynomial approximation of the graph Laplacian. In our baseline, we adopt a two-layer architecture with polynomial order 2 and a hidden dimension of 64. The model is trained for 10,000 epochs with a learning rate of 0.01 and a dropout rate of 0.5. By leveraging higher-order polynomial filters on either the symmetrically normalized or the random-walk normalized Laplacian, ChebNet enables effective aggregation of neighborhood information. Each layer is followed by nonlinear activation and dropout, which enhance the expressiveness of node representations.

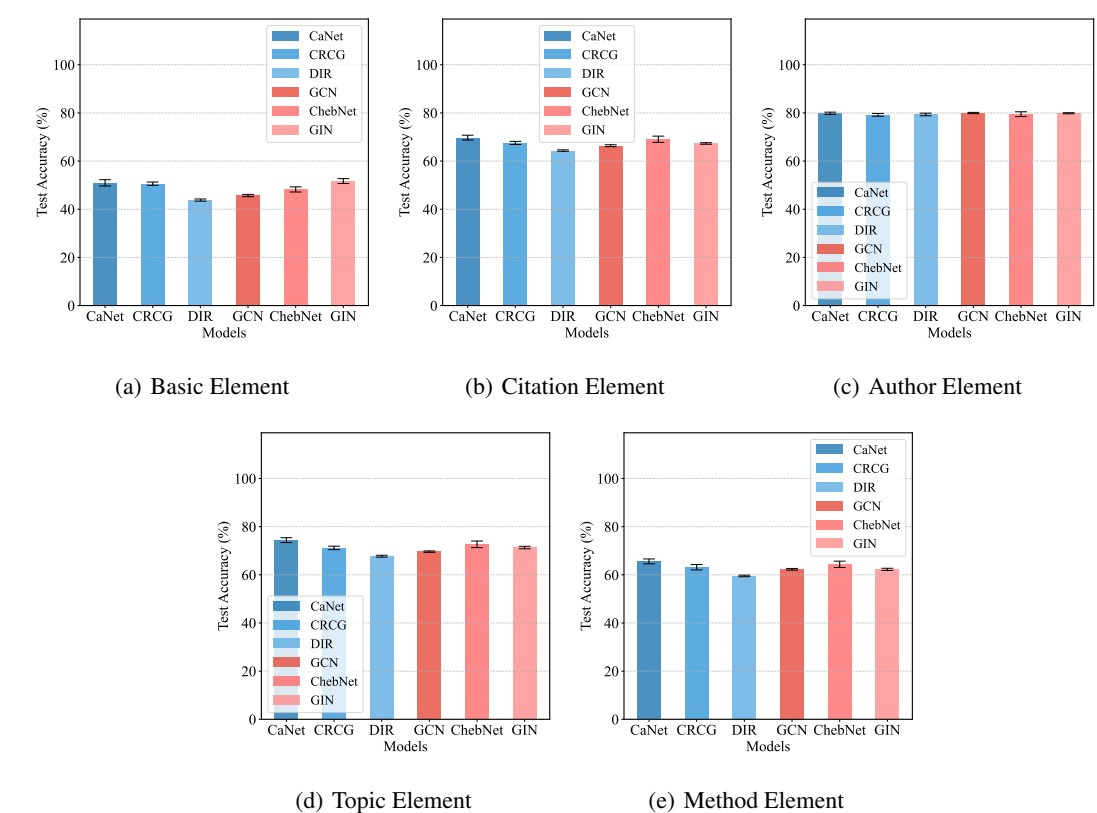

(a) Basic Element      (b) Citation Element      (c) Author Element

(d) Topic Element      (e) Method Element

Figure 14: Citation Confounders

**DIR.** This baseline is designed to capture causal structures within graphs. The model is configured with a hidden dimension of 128, a causal ratio of 0.7, a learning rate of 0.001, a batch size of 128, and is trained for 50 epochs. It estimates edge importance scores through convolutional encoding and a multilayer perceptron, and then separates subgraphs based on the causal ratio. The architecture incorporates three predictive branches, focusing on causal, confounding, and combined predictions. Training follows a dual-loss strategy with masking mechanisms, which emphasize causal signals while mitigating the influence of confounders.

**CRCG.** This baseline integrates causal representation learning into graph neural networks. With a hidden dimension of 32, a causal ratio of 0.25, a learning rate of 0.001, a batch size of 64, and 50 training epochs, this baseline jointly models causal and confounding structures. Edge importance is estimated by combining an encoder with a scoring mechanism, and graphs are split into causal and confounding subgraphs before being relabeled for consistency. The predictive module contains distinct branches for causal and confounding signals. Training employs a dual-loss scheme, and model performance is evaluated with multiple metrics, including accuracy, precision, and mean reciprocal rank.

**CaNet.** This baseline introduces a causal attention mechanism for robust graph learning. It is evaluated on the Citeseer dataset with a two-layer architecture, a hidden dimension of 64, three environments, a learning rate of 0.01, weight decay of $5 \times 10^{-4}$, and 40 training epochs. To ensure stability, experiments are repeated

three times. CaNet leverages Gumbel-Softmax to learn environment distributions and adopts a two-stage forward process, where the model outputs both predictions and regularization losses during training. A feature filtering module is applied at the input stage to suppress noisy or irrelevant features, and graph-level pooling is used for aggregation. This design strengthens the generalization ability of the model across different environments.

### F.3 DATASETS

**RWG-Molecular.** Each generated dataset contains 1900 graphs, among which 1500 graph samples are used as training samples, 200 samples as validation samples, and 200 samples as test samples. The number of nodes ranges from 50 to 80, the number of edges ranges from 60 to 120, the node feature dimension is 5, and there are 5 classes. In the experiment of Figure 3, the dataset is artificially synthesized molecular data, with a confounding ratio of 90% and an intervention probability of 100%. In the experiment of Figure 4, the dataset is artificially synthesized molecular data, with the confounding ratio ranging from 10% to 90%. In the experiment of Figure 5, the dataset is artificially synthesized molecular data with either a single large molecule block (index = 1, size = 50, branched = 10) or multiple small molecule blocks (indices = 1–10, size = 5, branches = 5), and a confounding ratio of 70% is applied. In the experiment of Figure 6, the training set of the dataset has a confounding ratio of 0%, while the validation and test sets have a confounding ratio of 70%. In the experiment of Table 2, the dataset is artificially synthesized molecular data, with a confounding ratio of 70%.

**RWG-Citation.** Each generated dataset contains 1900 graphs, among which 1500 graph samples are used for training, 200 samples for validation, and 200 samples for testing. The number of nodes ranges from 15 to 25, the number of edges ranges from 20 to 60, the node feature dimension is 5, and there are 5 classes. In the experiment of Figure 3, the dataset is artificially synthesized citation network data, with a confounding ratio of 90% and an intervention probability of 100%. In the experiment of Figure 4, the dataset is artificially synthesized citation network data, with the confounding ratio ranging from 10% to 90%. In the experiment of Figure 5, the dataset is artificially synthesized citation network data, incorporating mixed node information and complex structures (e.g., node relations), with a confounding ratio of 70%. In the experiment of Figure 6, the training set of the dataset is applied with a confounding ratio of 0%, while the validation and test sets are applied with a confounding ratio of 70%. In the experiment of Table 2, the dataset is artificially synthesized citation network data, with a confounding ratio of 70%.

**SPMotif.** In each generated dataset, there are 1900 graphs in total, with 1500 graphs used as training samples, 200 graphs as validation samples, and 200 graphs as test samples. The number of nodes ranges from 20 to 40, the number of edges ranges from 30 to 50, the node feature dimension is 5, and there are 5 classes. In the experiments of Figure 3, the dataset consists of synthetically generated primitive data, with a confounding ratio of 90% and an intervention probability of 100%. In the experiments of Figure 4, the dataset consists of synthetically generated primitive data, with the confounding ratio ranging from 10% to 90%. In the experiments of Figure 6, the training set of the dataset is generated with a confounding ratio of 0, while the validation and test sets are generated with a confounding ratio of 70%. In the experiments of Table 2, the dataset consists of synthetically generated primitive data, with a confounding ratio of 70%.

**CRCG.** This dataset is a synthetic graph classification dataset. Following the official setting, our generated data comprises 4,000 graphs in total, with 1,000 for training, 1,000 for validation, and 2,000 for testing. The dataset contains five classes, and the confounder ratio is set to 70%.

**CiteSeer.** This dataset is a citation network dataset consisting of 3,312 nodes, each with a 3,703-dimensional feature vector, and 4,723 edges. The dataset contains six classes.

**ENZYMES.** This dataset is a graph dataset constructed from protein tertiary structures. It contains 600 graphs with a total of 19,580 nodes and 174,564 edges. Each node has a 3-dimensional feature vector, and the dataset covers six classes.

