# OpenReview forum: "A Closer Look at the Application of Causal Inference in Graph Representation Learning"
_ICLR.cc/2026/Conference — ICLR 2026 Conference Withdrawn Submission_

### Official Review · Reviewer_47W9 · 2025-10-28

**Soundness:** 3
**Presentation:** 1
**Contribution:** 2
**Rating:** 4
**Confidence:** 4

**Summary:**

This paper provides a critical examination of causal inference in graph representation learning. The authors argue that common practices, like merging nodes and edges into single causal variables, can violate the core assumptions of causal inference. They propose a theoretical model based on the smallest graph elements to ensure validity, analyzing the high cost of perfect causal modeling and the strict conditions needed for simplification. To support their theory, they introduce a realistic synthetic dataset and a plug-and-play module that reduces data complexity, showing consistent improvements.

**Strengths:**

1. The paper presents a rigorous and well-justified theoretical analysis that clearly explains why naive variable merging can undermine the causal validity of graph representation learning.

2. By grounding the analysis in the smallest indivisible causal units of graph data, the paper establishes a coherent and insightful theoretical framework for causal modeling in GNNs, with each theoretical component logically building upon the previous one.

3. The proposed REC module is lightweight, broadly applicable, and empirically effective, demonstrating consistent performance gains across diverse graph learning architectures.

**Weaknesses:**

1. The paper’s clarity could be improved. For instance, Line 58 states “We provide an intuitive example in Figure 1,” yet the example is not further elaborated upon, and readers are instead referred to Proposition 1 for details—making it less intuitive. Either a clearer explanation of the example or its removal from the Introduction would enhance readability. In addition, Lines 67–84 are somewhat repetitive, as the listed contributions overlap substantially with those in Lines 67–74. Streamlining these sections would make the presentation more concise.

2. The connection between Proposition 5 and the proposed REC module is not clearly established. It would strengthen the paper to provide theoretical or empirical evidence that REC indeed achieves the goal articulated in Proposition 5.

3. The experiments in Table 2 lack comparisons with other causal inference baselines. Including such baselines would better demonstrate the advantages of the proposed method and validate its causal modeling effectiveness.

**Questions:**

1. What are the differences between REC and gated neural networks?

2. How does the $\gamma$ in Eq.(3) influence model performance?

---

### Official Review · Reviewer_sL8t · 2025-10-28

**Soundness:** 4
**Presentation:** 4
**Contribution:** 4
**Rating:** 6
**Confidence:** 4

**Summary:**

This paper addresses an important but under-discussed theoretical issue in causal graph representation learning (GRL): that aggregating multiple graph components into coarse causal variables can break the Causal Markov and Causal Faithfulness assumptions fundamental to causal inference.
The authors formalize this violation (Proposition 1) and propose a structural causal model defined over atomic graph elements. They then introduce a practical mitigation mechanism, REC (Redundancy Elimination Component), which learns to prune redundant or confounding features in GNNs. Experiments on synthetic and benchmark datasets show consistent performance gains.

**Strengths:**

1. Timely theoretical framing. The observation that naive aggregation undermines causal validity is central yet rarely formalized in GRL.
1. Clear theoretical–practical bridge. REC operationalizes the theoretical insight by providing a differentiable gating mechanism that reduces spurious dependencies.
1. Readable and coherent. The exposition is clear and the argument logically structured from theory to implementation.
1. Empirical support. Experiments are well-organized and reproducible, with gains that, while moderate, are consistent across datasets.
1. Potential for broader impact. The notion of “aggregation safety” could influence future work on causal GNN design and representation learning.

**Weaknesses:**

1. Background context. The key theoretical insight (aggregation may violate Markov and Faithfulness) is not new. The paper cites Spirtes (2009) and Pearl (2009) but omits earlier work on causal abstraction and macro-variable formation (e.g., Beckers & Halpern 2018/2019; Chalupka et al. 2016–2017). Including this would clarify that the contribution lies in the response (REC), not the observation itself.
1. Proof and notation clarity. Theorem 3’s constants (λ, r(Mmicro), etc.) are undefined, making interpretation difficult.
1. Scope of validation. REC is evaluated by predictive accuracy; causal-specific metrics (e.g., independence preservation, reduction of spurious links) would better connect to the paper’s motivation.
1. Positioning relative to prior causal-GNN work. Comparing REC to causal-attention or invariant-representation methods (e.g., Sui 2022; Chen 2022) would help establish distinctiveness.

**Questions:**

Suggestions:

1. Cite and briefly discuss the causal-abstraction literature (Beckers & Halpern; Chalupka et al.) to contextualize Proposition 1.
1. Clarify or illustrate constants in Theorem 3 with intuition or examples.
1. Add causal diagnostics—e.g., tests of conditional independence before/after REC—to show that REC improves causal fidelity, not just accuracy.
1. Explicitly contrast REC with prior “causal attention” or “invariant GNN” modules, emphasizing its aggregation-safety motivation.
1. Discuss whether REC approximates a form of valid macro-abstraction, connecting to causal representation learning theory.

---

### Official Review · Reviewer_rGWT · 2025-10-30

**Soundness:** 2
**Presentation:** 3
**Contribution:** 2
**Rating:** 2
**Confidence:** 3

**Summary:**

The paper addresses the modeling of causal relations in graph representation learning. The authors present a theoretical framework for making causally informed predictions in the case of graph representation learning. They propose merging some variables into a single variable, which can allow simplifying the model and make learning it easier, and rightfully point out that this merging is non-trivial from a causal perspective.
They propose a new, more flexible benchmark for causal graph representation learning. In an example study, they demonstrate how current methods can fail by evaluating multiple methods across different settings, which do not appear over-engineered to be adversarial. They propose their own modification that can be incorporated into existing methods and show empirically that their method is beneficial in the proposed settings.

**Strengths:**

* The paper has a good motivation for using causality in the proposed setting and for merging variables.

* The authors  propose a highly parameterized framework motivated by previous benchmarks and limited parameterization in existing work.

* The authors  use multiple GNN models - both vanilla GNNs  and causally enhanced versions.

* Through thoughtfully crafted examples, they show how methods can fail in some settings when there are confounders and how interventions can help. The settings do not appear to be overly adversarial and are well grounded in previous work.

* The proposed method is very flexible and can be applied to a broad set of models.

* The results show improvement across all models.

**Weaknesses:**

* *Theorem 2* - In your setting, you make an assumption that exogenous variables U can be connected to more than one X. In such a setting, since we do not observe U variables, we have unobserved confounders. The PC method you use in the proof assumes no hidden confounding. As far as I know, in such cases we are recovering a PAG (partial ancestral graph) (see: Spirtes, P., Meek, C., & Richardson, T. (1995, August). Causal inference in the presence of latent variables and selection bias. In Proceedings of the Eleventh conference on Uncertainty in artificial intelligence (pp. 499-506)). Also, in the proof of Theorem 2, I do not see any explanation of how you discover and distinguish between $X_{cfd}$ and $X_{asoc}$—these steps are not explained.


* *Theorem 3* - As motivation for Theorem 3, you write "achieve perfectly accurate causal modeling using a GNN model"—however, it is not clear to me what you mean by this perfect modeling. Is it knowing the full causal graph? Subsequently, I do not understand what this number of interventions is required for (to recover single true graph?). Also, I do not understand the setting in which you are doing the interventions -- "λ denotes the average times that each variable occurs among each of the samples within dataset G"—what is the meaning of λ, and do we assume that we do not observe all variables for each datapoint? Additionally, the theoretical results that are cited, for example Choo, Davin, Kirankumar Shiragur, and Arnab Bhattacharyya, "Verification and search algorithms for causal DAGs," Advances in Neural Information Processing Systems 35 (2022): 12787-12799, are about recovering DAG G from an essential graph of G, so the input has to be MEC (Markov Equivalence Class), I do not know where this MEC comes from (see also points above).

* *Theorem 5* - Can you explain what you mean by "precisely models the causal mechanism"? Do you assume that you model correctly the deterministic functions of $Y = f($X_{caus}$)$? I do not see how this assumptions leads to this conclusion.

* *REC method* - I do not clearly understand the connection of REC to the theorems apart from Theorem 2, from which I understand that a subset of variables $X_{cause}$ is sufficient for predicting Y. So is it a method of feature selection? Is there any evidence that we see variables merging? I do not see that it is a causal way of doing it—do we not simply make the model use fewer variables (which is a good thing if we know that not all variables are important)?

**Questions:**

* Do you assume that Y is a leaf and a deterministic function of some subset X?

* Is it easy to pick hyperparameters of REC, and do they behave stably? Also, if we have a label that is dependent on substantially more or fewer Xs, do the parameters need to change? Is the number of eliminated variables constant across datasets?

* Did you consider comparing the improvement of REC in a setting with interventions? Can it make explicit use of interventional data?

---

### Official Review · Reviewer_Eye7 · 2025-11-01

**Soundness:** 1
**Presentation:** 2
**Contribution:** 2
**Rating:** 2
**Confidence:** 2

**Summary:**

** Summary

This paper studies the graph based prediction tasks where simplifying the graph (typically by merging nodes) may lead to harms to prediction accuracy. The work focuses on formalising the graph as a causal graph, and claims that merging nodes that fail to follow certain causal structure criteria may cause failure in identifying the correct causal structure, consequently leading to inaccurate predictions. The work proposes theoretical analysis on criteria of merging nodes, so to maintain the core causal structure identifiable. Then, it proposes an algorithm that merges node following their criteria, and shows the method’s efficacy by experiments on synthetic data.

** Recommendation

I would like to recommend a rejection to this paper for its soundness on theories of causal discovery. Generally, the problem is interesting and I can see the strong connection to causality research, especially the identifiability of SCM and proxy of covariates. However, I found the paper fails to formalise the problem clearly (especially for non-GNN focused researchers like me), and some theoretical parts may be problematic. I am not an expert of GNN related areas, and just gave my recommendation from the angle of causality.

**Strengths:**

1. The paper raises an interesting question, which has a clear link to causality theories.
2. The proposed node merging strategy can be effective though the theoretical analysis may be problematic. This is also proved by the experiment results.

**Weaknesses:**

1. My main concern is the theoretical part of causality. The authors need to be clear about the causality theories and make their statements clear. For instance, in the proof of Proposition 1, the author try to show a contradiction against the local Markov property by giving a claim: in structure S_i -> S_j <- S_k, S_j and S_k should be independent conditioned on S_i, but this is not true. However, this is different from the local Markov property, which says variable v is independent from any variable that is not a descendent or parent of v conditioned on v’s parents, but this is not applicable to the example in the proof. A similar problem exists in Theorem 2’s proof (Line 846 - 848), where b may be a descendent of a.
2. The causality related definitions need to be revised. For instance, a SCM clearly specifies the structural functions, but in this work, it seems the model is more like a graphical causal model. Another instance is Line 161-162. My understanding is that the target is to learn the causal relationship between X and Y, so X^{cfd} may be different from the term “confounder” in causality.
3. Theorem 4 needs further clarification. If we merge some nodes from group 2 and 3, and apply joint intervention on the group, we may still identify the causal relationship between group 3 and Y. Please correct me if I am wrong on this.
4. The problem definition needs to be clear. My feeling is the ultimate target is to predict an accurate Y, but this is not specified in the paper. In the experiment, the authors mention the “performance” of the methods, but it will improve the readability by make such concepts clear.
5. The authors may improve the notations to improve the readability. For instance, use capital letters to denote variables, small letters to denote values, and bold capital letters to denote variable sets.

**Questions:**

See weakness comments.

---

### Note · Authors · 2025-12-08

I have read and agree with the venue's withdrawal policy on behalf of myself and my co-authors.